# High-rate and selective conversion of $CO_2$ from aqueous solutions to hydrocarbons

Cornelius A. Obasanjo[1,3], Guorui Gao[1,3], Jackson Crane [1,3], Viktoria Golovanova[2], F. Pelayo García de Arquer [2] & Cao-Thang Dinh [1] ✉

Electrochemical carbon dioxide ($CO_2$) conversion to hydrocarbon fuels, such as methane ($CH_4$), offers a promising solution for the long-term and large-scale storage of renewable electricity. To enable this technology, $CO_2$-to-$CH_4$ conversion must achieve high selectivity and energy efficiency at high currents. Here, we report an electrochemical conversion system that features proton-bicarbonate-$CO_2$ mass transport management coupled with an in-situ copper (Cu) activation strategy to achieve high $CH_4$ selectivity at high currents. We find that open matrix Cu electrodes sustain sufficient local $CO_2$ concentration by combining both dissolved $CO_2$ and in-situ generated $CO_2$ from the bicarbonate. In-situ Cu activation through alternating current operation renders and maintains the catalyst highly selective towards $CH_4$. The combination of these strategies leads to $CH_4$ Faradaic efficiencies of over 70% in a wide current density range (100 – 750 mA cm$^{-2}$) that is stable for at least 12 h at a current density of 500 mA cm$^{-2}$. The system also delivers a $CH_4$ concentration of 23.5% in the gas product stream.

Electrochemical carbon dioxide ($CO_2$) reduction (ECR) to chemical fuels offers a large-scale solution to the long-term storage of renewable electricity[1]. Among many possible ECR products, methane ($CH_4$) is an appealing target for energy storage applications because of its high energy density (55.5 MJ kg$^{-1}$), widespread use, and large market size (responsible for ~23% of the global energy use)[2]. $CH_4$ produced from $CO_2$ and renewable electricity could be readily integrated into the existing natural gas infrastructure, providing a direct path to its decarbonization[3].

Practical ECR systems must operate at high current densities (often >300 mA cm$^{-2}$) with high selectivity and energy efficiency[1,4]. To reduce separation costs, high product concentrations and avoiding $CO_2$ loss to carbonate crossover are desirable[5]. Gas-phase ECR systems, including both flow-cell and membrane electrode assembly (MEA) configurations, have been used as platforms to achieve selective $CO_2$ conversion at high current densities[6–10]. Using gas-phase systems, $CO_2$-to-$CH_4$ conversion with high Faradaic efficiencies (FEs) (60–70%) at relatively high current densities (200–300 mA cm$^{-2}$) have been

demonstrated[11–22]. However, gas-phase systems often use alkaline electrolytes or anion exchange membranes to achieve these high selectivities. In these configurations, carbonate formation (with alkaline electrolytes) or crossover (with anion exchange membranes) is significant, requiring additional energy to recycle the electrolyte or $CO_2$[5,23,24].

Carbonate and bicarbonate-fed systems have recently been developed to integrate $CO_2$ capture and conversion steps within a single system[25–28]. In this architecture, $CO_2$ is generated in situ inside the reactor when protons generated from a bipolar membrane (BPM) react with (bi)carbonate. This system offers effective carbon utilization that bypasses the energy-intensive step of extracting $CO_2$ from a $CO_2$ captured solution[25,29]. It also allows the production of gas products with high concentrations, as the gaseous products do not mix with the $CO_2$ feedstock[29]. Using (bi)carbonate fed systems, high carbon monoxide (CO) and formate selectivity (FEs over 70%) at relatively high partial current densities (70–150 mA cm$^{-2}$) have been demonstrated[25,26,28,30].

[1]Department of Chemical Engineering, Queen's University, Kingston, ON K7L 3N6, Canada. [2]ICFO–Institut de Ciències Fotòniques, The Barcelona Institute of Science and Technology, Barcelona 08860, Spain. [3]These authors contributed equally: Cornelius A. Obasanjo, Guorui Gao, Jackson Crane. ✉e-mail: caothang.dinh@queensu.ca

Selectivity for $CH_4$ production from recent bicarbonate-fed systems is relatively low compared to gas-phase systems, however. The current state-of-the-art for $CH_4$ production from bicarbonate exhibited a FE of 30% and a partial current density of 120 mA cm$^{-2}$ ref. [27]. We hypothesized that this limited performance stems from a yet uncontrolled reaction environment, including both local $CO_2$ concentration and local pH, and the lack of selective catalysts, possibly because Cu reconstruction at high current density typically favors the hydrogen evolution reaction[31–34].

In this work, we unveil the critical role of electrode pore size on local $CO_2$ availability in bicarbonate-fed systems using BPMs. We found that large Cu pores enhance the transport of dissolved $CO_2$ and favor bicarbonate conversion into $CO_2$ leading to high local $CO_2$ concentration. We further develop an in-situ catalyst activation strategy that generates and maintains highly specific Cu surfaces for selective $CH_4$ production from $CO_2$ dissolved in aqueous solutions. We implement this catalyst in large pore electrodes to report selective and concentrated $CH_4$ generation, at high current densities. Our aqueous-fed system achieves a $CO_2$-to-$CH_4$ conversion with over 70% FE at a wide current density range of 100–750 mA cm$^{-2}$, with a record-high $CH_4$ partial current density of over 500 mA cm$^{-2}$. The system is also stable, maintaining its high $CH_4$ FE for at least 12 h at a current density of 250–500 mA cm$^{-2}$. Our aqueous-fed system also outperforms previous systems in terms of full-cell energy efficiency and delivers a record $CH_4$ concentration of up to 23.5% in the gas outlet stream.

## Results

### Modeling the reaction environment and $CO_2$ availability

Electrochemical $CO_2$ reduction relies on high $CO_2$ availability within the catalyst domain. To study the role of $CO_2$ availability in a bicarbonate-fed ECR system, we used one-dimensional multiphysics modeling. The cation exchange layer and porous copper catalyst subsystems are considered in the model (Fig. 1a, b). Key physics are considered including species and charge transport, electrocatalytic reactions, $CO_2$ phase transfer, and buffer equilibrium reactions, with constants derived from past work[35–39]. A proton flux from water dissociation within the BPM, proportional to the total integrated current density, is imposed on the anion exchange layer/cation exchange layer interface. A mass flux boundary condition is imposed on the catalyst layer/flow channel interface from the bulk electrolyte. In this system, $CO_2$ comes from two sources: dissolved $CO_2$ in the $KHCO_3$ electrolyte and $CO_2$ generated during the reaction. Under reverse bias, protons ($H^+$) generated from the surface of the BPM react with bicarbonate ions ($HCO_3^-$) in the electrolyte, forming $CO_2$ in the solution.

$$HCO_3^- + H^+ \rightarrow CO_2 + H_2O \quad (1)$$

The $CO_2$ formed in situ diffuses to the surface of the catalyst, and when formed in excess of the solubility limit, forms bubbles on the surface of the membrane (Fig. 1b). $CH_4$ is generated in an 8-electron reaction, consuming $CO_2$ and producing $OH^-$.

$$CO_2 + 4H_2O + 8e^- \rightarrow CH_4 + 8OH^- \quad (2)$$

State-of-the-art bicarbonate electrolyzers rely on catalysts within a dense porous matrix with the $CO_2$ source in-situ generation from bicarbonate[26–28].

Due to the critical importance of $CO_2$ availability on electrolyzer performance[40], we examined $CO_2$ availability in the catalyst domain for a

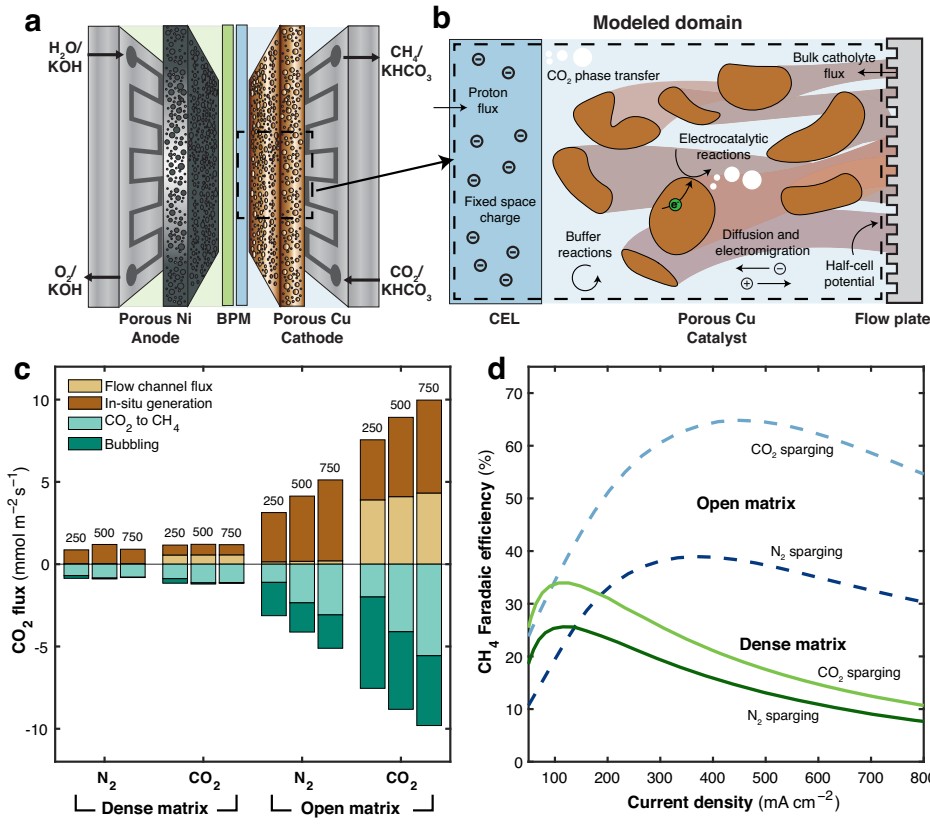

**Fig. 1 | Modeling reaction environment and $CO_2$ availability. a** Schematic illustration of the aqueous solution-fed ECR system using porous Cu cathode, Ni foam anode and bipolar membrane (BPM). **b** Schematic of modeled domain with key physics annotated. **c** $CO_2$ flux components at three current densities, 250, 500, and 750 mA cm$^{-2}$ for dense matrix and open matrix catalysts with $N_2$ sparging or $CO_2$ sparging. 0.3 M $KHCO_3$ was used as an electrolyte for all modeling. **d** Modeled $CH_4$ FE as a function of current density for a dense matrix catalyst (solid lines) and open matrix catalyst (dashed lines) with $CO_2$ sparging and $N_2$ sparging.

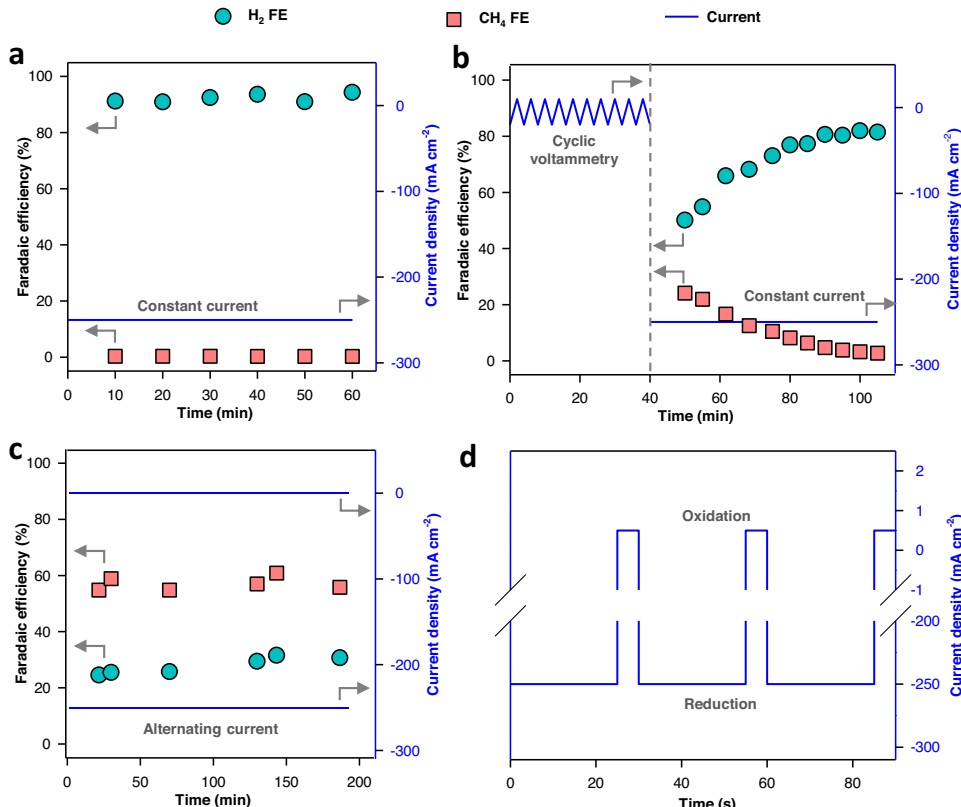

**Fig. 2 | Electrocatalytic CO₂ reduction with constant and alternating current operations. a** Product distribution of Cu mesh over time operated at a constant reduction current density of 250 mA cm⁻². **b** Product distribution of Cu mesh after catalyst surface activation using cyclic voltammetry (CV) between −20 and 10 mA cm⁻² for 10 cycles, followed by operation at a constant reduction current density of 250 mA cm⁻². **c** Product distribution of Cu mesh operated using alternating negative and positive currents (oxidation current density of 0.5 mA cm⁻² and oxidation time of 3 s; reduction time of 25 s and CO₂ reduction current of 250 mA cm⁻². **d** Magnification in time of the square-wave alternating reduction and oxidation current density in (**c**).

dense porous matrix catalyst. We found that most CO₂ in-situ generation via Eq. 1 occurs near the catalyst-membrane interface (Fig. S1), while dissolved CO₂ from the bulk electrolyte enters via the flow channel. To generate substantial quantities of C₁₊ products via ECR, CO₂ must readily diffuse from the boundaries into the catalyst domain. The model is used to perform a detailed CO₂ accounting at different current densities to understand CO₂ dynamics within the system. CO₂ fluxes are sub-divided into four components: (1) flux from the flow channel, (2) in-situ generation (although CO₂ can also be consumed, on net, via fixing into (bi) carbonate), (3) conversion of CO₂ to CH₄, and (4) aqueous to vapor phase transfer advected to the flow channel via bubbling. In the dense matrix catalysts with N₂ sparging at high current density, in-situ CO₂ generation is severely restricted, while flow-channel flux is nearly non-existent (Fig. 1c), leading to low CH₄ selectivity (Fig. 1d).

Given the dominant role of CO₂ diffusivity on ECR rates, we hypothesized that an open matrix design, which leads to higher diffusion and boundary mass transfer, would improve performance. Modeling suggests that when switching from a dense matrix to an open matrix, both with N₂ sparging, performance at high current densities improves substantially (Fig. 1d). For example, at 500 mA cm⁻² total current density, CH₄ FE increases from 13% to 37% when switching from the dense to the open matrix catalyst. Mechanistically, modeling suggests that high diffusivity from the open matrix facilitates much higher in-situ CO₂ generation (Fig. 1c). The CO₂ transport from the flow channel in the open matrix N₂ sparging case remains, however, low. This leads to adequate CO₂ availability on the membrane side of the catalyst domain, but poor CO₂ availability on the flow-channel side (Fig. S2). We further hypothesized that it would be possible to improve the utilization of the entire catalyst domain by

increasing CO₂ concentration from the flow channel by sparging the electrolyte with CO₂ instead of N₂. CO₂ flux tracking shows that when sparging with CO₂, the CO₂ influx from the flow channel improves substantially, leading to CH₄ FEs of up to 65% at 500 mA cm⁻². Meanwhile, CO₂ sparging makes only a minor difference in the dense matrix cases (Fig. 1d) presumably because the low diffusivity and mass transport prevent CO₂ from adequately penetrating the domain. Similar trends are observed for other catholyte concentrations from 0.1 to 1 M KHCO₃ (Fig. S3).

These results suggest a path forward and design principles to realize bicarbonate-fed electrolyzers with increased performance: a highly porous matrix enables efficient utilization of both dissolved CO₂ from electrolyte and in-situ formed CO₂ from bicarbonate.

## In-situ generation of selective catalysts

To implement the open matrix strategy, we selected a cathode consisting of a copper mesh with large pores (average pore diameter of 150 μm and average Cu wire of 100 μm) and a pore density of 100 pores per inch (Fig. S4). Electrochemical CO₂ reduction was performed using an aqueous-fed system coupled with a BPM operated in reverse bias mode (Fig. S5). A 0.3 M KHCO₃ solution saturated with CO₂ was used as both the electrolyte and CO₂ source and a 1 M KOH solution was used as the anolyte (Fig. 1a).

We first performed the electrochemical reaction using as-received Cu mesh at a current density of 250 mA cm⁻². In this configuration, H₂ is the major product with Faradaic Efficiency (FE) over 90% throughout the test while the FE for CH₄ is less than 0.5% (Fig. 2a).

To improve CO₂ reduction selectivity and to suppress H₂ evolution reaction, we sought to reconstruct the surface of Cu catalyst using an electrochemical oxidation−reduction process to form Cu

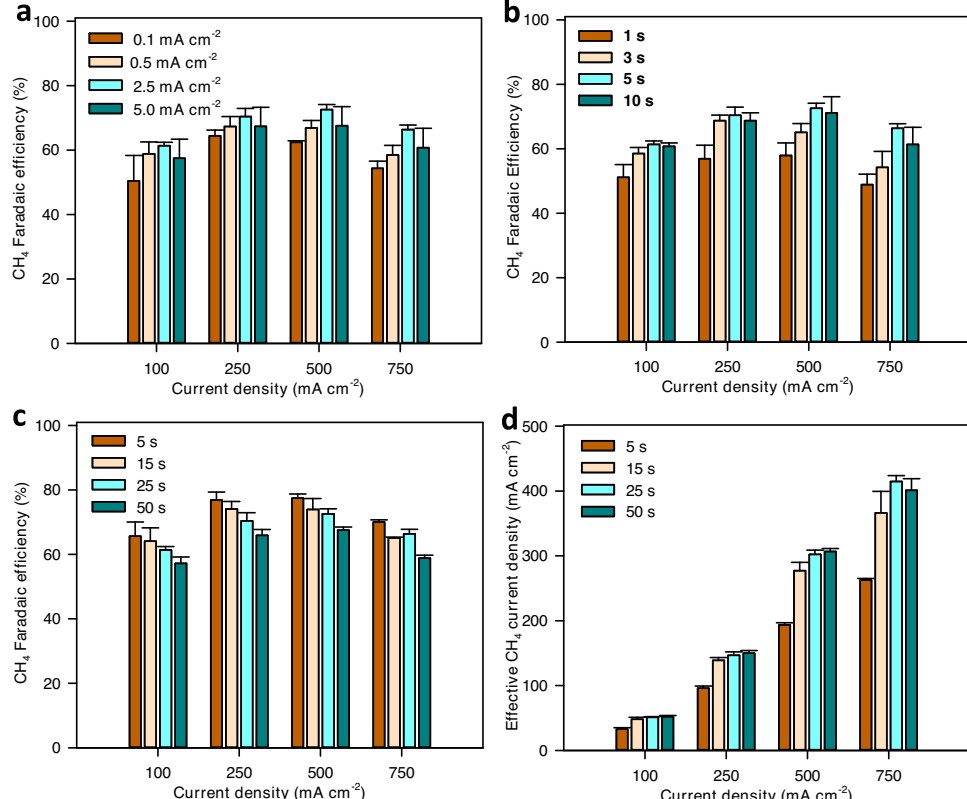

**Fig. 3 | Optimization of oxidation and reduction conditions. a** Effect of oxidation current on CH$_4$ selectivity. Oxidation and reduction times were fixed as 5 s and 25 s, respectively. **b** Effect of oxidation time. The oxidation current density was fixed at 2.5 mA cm$^{-2}$ and the reduction time was 25 s. **c** Effect of reduction time. Oxidation current density and time were fixed as 2.5 mA cm$^{-2}$ and 5 s, respectively. **d** Effect of reduction time on effective CH$_4$ partial current density. The effective CH$_4$ current was calculated as follows: effective current = (total current) × (Faradaic efficiency) × (time for reduction cycle)/(time for reduction cycle + time for oxidation cycle). Data for FE of other gas products are shown in Figs. S8–S10. The error bars represent SD ($n$ = 3 independent replicates).

oxide-derived catalysts, which have been found to be selective catalysts for CO$_2$ reduction to hydrocarbons[41–45]. To form the Cu-oxide-derived catalyst, we performed repeated cyclic voltammetry (10 cycles) in the current density range of −20 to +10 mA cm$^{-2}$ before performing CO$_2$ reduction (Fig. 2b). We found that the CH$_4$ FE increased to 25% while the H$_2$ FE decreased to 50% when Cu surface is reconstructed via reduction–oxidation cycling (Fig. 2b). However, the CH$_4$ FE rapidly decreased to 3% after 60 min of reduction time. These results suggest that the oxide-derived Cu surface can be selective but unstable for CH$_4$ formation.

From the results above, we reasoned that periodic Cu surface reconstruction would enable selective and stable CH$_4$ formation. To this end, we applied a square-wave alternating current in which the electrical current alternated between a negative and a positive current. Using this operation mode, the catalyst is periodically oxidized during CO$_2$ reduction (Fig. 2c, d). The oxidation current density and time were fixed at 0.5 mA cm$^{-2}$ and 3 s, respectively, and the reduction time was 25 s. We found that, under these conditions, CH$_4$ was the major product with an FE of 55–60% maintained throughout the reaction. Meanwhile, H$_2$ formation is suppressed, with a stable FE around 30% (Fig. 2c). Other minor products include C$_2$H$_4$ (1–3%) and CO (<1%). During the oxidation cycle, H$_2$ produced from the reduction cycle may be oxidized, contributing to the observed low H$_2$ FE. However, because the number of charges during the oxidation cycle (0.0015 C cm$^{-2}$) is much smaller than those in the reduction reaction (6.25 C cm$^{-2}$), this possible contribution is insignificant.

## Optimizing oxidation and reduction conditions
Having identified a strategy for the selective production of CH$_4$, we sought to further optimize the oxidation-induced Cu surface

modification by varying the oxidation and reduction current and time. We first studied the effect of oxidation current on CH$_4$ selectivity. The oxidation time was fixed at 5 s while we varied the oxidation current density between 0.1 and 5 mA cm$^{-2}$. The reduction time was fixed at 25 s and the reduction current density varied in the range of 100–750 mA cm$^{-2}$. We found that higher oxidation current density generally improves CH$_4$ selectivity (Fig. 3a). Particularly, with an oxidation current density of 2.5 mA cm$^{-2}$, CH$_4$ FE of over 70% was achieved at 250 mA cm$^{-2}$ and 500 mA cm$^{-2}$. At a current density of 750 mA cm$^{-2}$, we achieved a CH$_4$ FE of 67.8%, corresponding to a partial CH$_4$ current density of 508.5 mA cm$^{-2}$. C$_2$H$_4$ is the other main product with an FE in the range of 2–6%. The total FE for liquid products is less than 5% (Fig. S6). The full cell voltage was 3.52 V at a current density of 100 mA cm$^{-2}$ and increased to 4.25 V, 5.4 V, and 7.57 V as the current density increased to 250 mA cm$^{-2}$, 500 mA cm$^{-2}$, and 750 mA cm$^{-2}$, respectively (Figs. S6, S7). When the oxidation current density is higher than 2.5 mA cm$^{-2}$, further increases in oxidation current do not improve CH$_4$ selectivity (Fig. 3a).

To study the effect of oxidation time, we fixed the oxidation current density at 2.5 mA cm$^{-2}$ and varied the oxidation time in the range of 1 to 10 s. As with oxidation current, high oxidation time favors CH$_4$ production, especially at high current densities (Fig. 3b). To investigate the effect of reduction time, we fixed oxidation current density and time at 2.5 mA cm$^{-2}$ and 5 s, respectively, and varied the reduction time in the range of 5 to 50 s. Reducing the reduction time increases CH$_4$ selectivity substantially (Fig. 3c). With a reduction time of 5 s, CH$_4$ FE higher than 70% was achieved at all current densities in the range of 100–750 mA cm$^{-2}$. Notably, CH$_4$ FE reached 77% at 500 mA cm$^{-2}$ and a CH$_4$ partial current of 525 mA cm$^{-2}$ was achieved at current density of 750 mA cm$^{-2}$. We reason that highly selective Cu

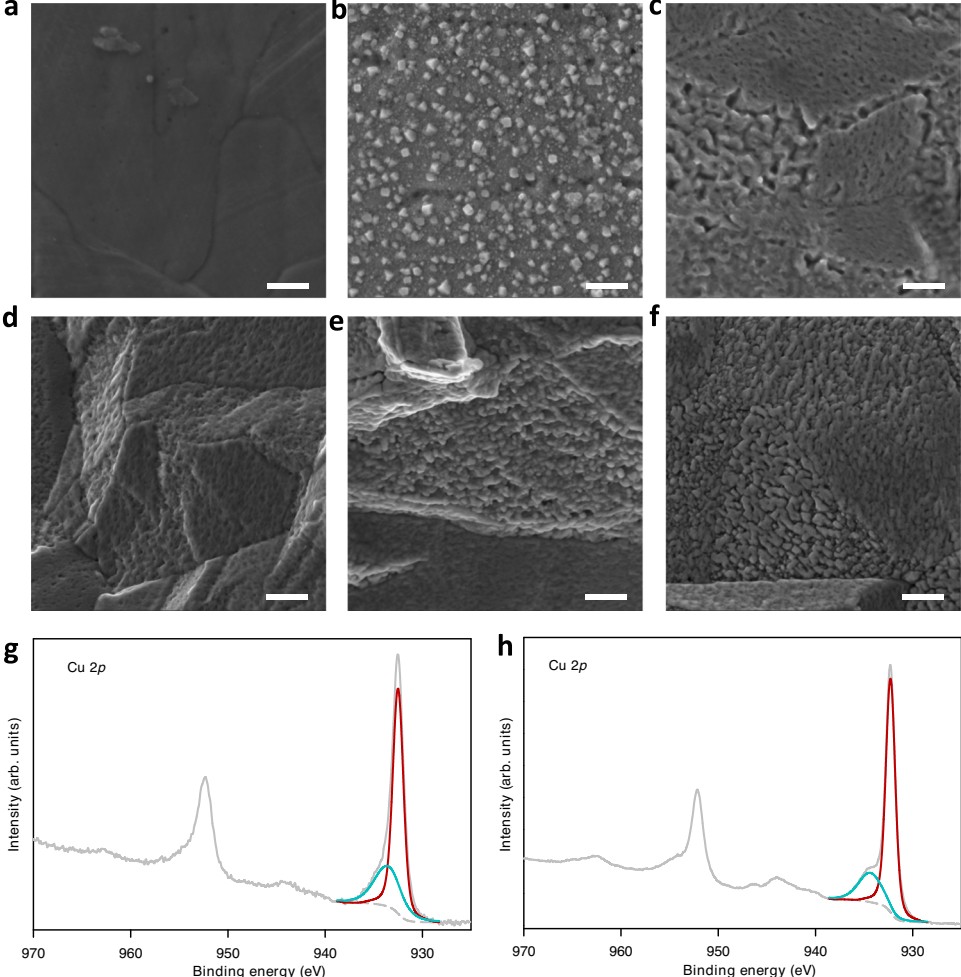

**Fig. 4 | Surface changes induced by alternating reduction–oxidation current.**
**a, b** Scanning electron microscopy (SEM) images of Cu mesh electrode before (**a**) and after (**b**) $CO_2$ reduction using fixed reduction current. Samples SEM image after electrolysis with alternating reduction–oxidation times of 25 s–5 s (**c**), 50 s–5 s (**d**), 15 s–5 s (**e**), and 5 s–5 s (**f**). Cu 2$p$ X-ray photoelectron spectroscopy (XPS) spectra of Cu mesh electrode after $CO_2$ reduction using fixed currents (**g**); and alternating 25 s–5 s reduction–oxidation current (**h**). The samples for SEM and XPS were collected after being tested at 100, 250, 500, and 750 mA cm$^{-2}$ reduction current densities for 40 min at each current regime (total reaction time of 160 min). The oxidation current density and time were 2.5 mA cm$^{-2}$ and 5 s, respectively. The reduction time considered for samples (**c**)–(**f**) are 25 s, 50 s, 15 s, and 5 s, respectively. Scale bar in Fig. **a**–**f** = 200 nm.

species are generated during the oxidation step. During the reduction cycle, the Cu surface is gradually reconstructed leading to a decrease in CH$_4$ selectivity. Thus, shortening reduction time enables the efficient use of highly selective Cu species, leading to higher CH$_4$ selectivity. It is also possible that as the reduction timescale becomes sufficiently short, replenished $CO_2$ within the catalyst domain enables higher CH$_4$ selectivity.

While decreasing reduction time increases instantaneous CH$_4$ selectivity, it also reduces the effective duty cycle for $CO_2$ reduction (reduction time/(reduction time + oxidation time)). To evaluate the effective production rate of CH$_4$, we normalized CH$_4$ partial current density to total operation time (reduction and oxidation time). The highest effective CH$_4$ partial current recorded was over 400 mA cm$^{-2}$, achieved with a reduction time of 25 s (Fig. 3d). Longer reduction times lower CH$_4$ selectivity while shorter reduction times reduce effective operating time. Our results show a tradeoff between CH$_4$ selectivity and effective partial current density towards reduction time.

**Catalyst surface changes induced by oxidation**
To understand the effect of different operating procedures on the surface changes of the catalysts, we performed scanning electron microscopy (SEM) characterization of the catalyst before and after $CO_2$

reduction reactions. We found that $CO_2$ reduction conditions have a significant impact on the surface of Cu catalysts. SEM characterizations after ECR reactions with constant reduction current reveal the formation of Cu nanoparticles with irregular shapes (Figs. 4b, S12, S13) as opposed to a smooth surface of the sample before the reaction (Figs. 4a, S11). This observation is consistent across different locations of multiple samples. Previous studies using operando characterizations of catalysts under ECR conditions suggested that the Cu surface is dynamic and undergoes significant morphological changes due to multiple processes, including dissolution/redeposition, agglomeration, fragmentation and reshaping[33,34,46–48]. Formation of Cu nanoparticles on the Cu surface during ECR has been observed in a previous study at a current density below 50 mA cm$^{-2}$ ref. [12]. Because the surface changes are accelerated at higher reaction rates or more negative applied potentials[33], formation of Cu nanoparticles at current densities of 500 to 750 mA cm$^{-2}$ in our system is reasonable. In sharp contrast to the constant reduction operation, alternating reduction–oxidation currents were found to induce the formation of a porous Cu surface (Fig. 4c). The porous structure becomes more pronounced when the reduction time is reduced from 50 s to 5 s (Figs. 4d–f, S14), suggesting that oxidation cycle frequency is an important factor governing the morphology of the catalyst.

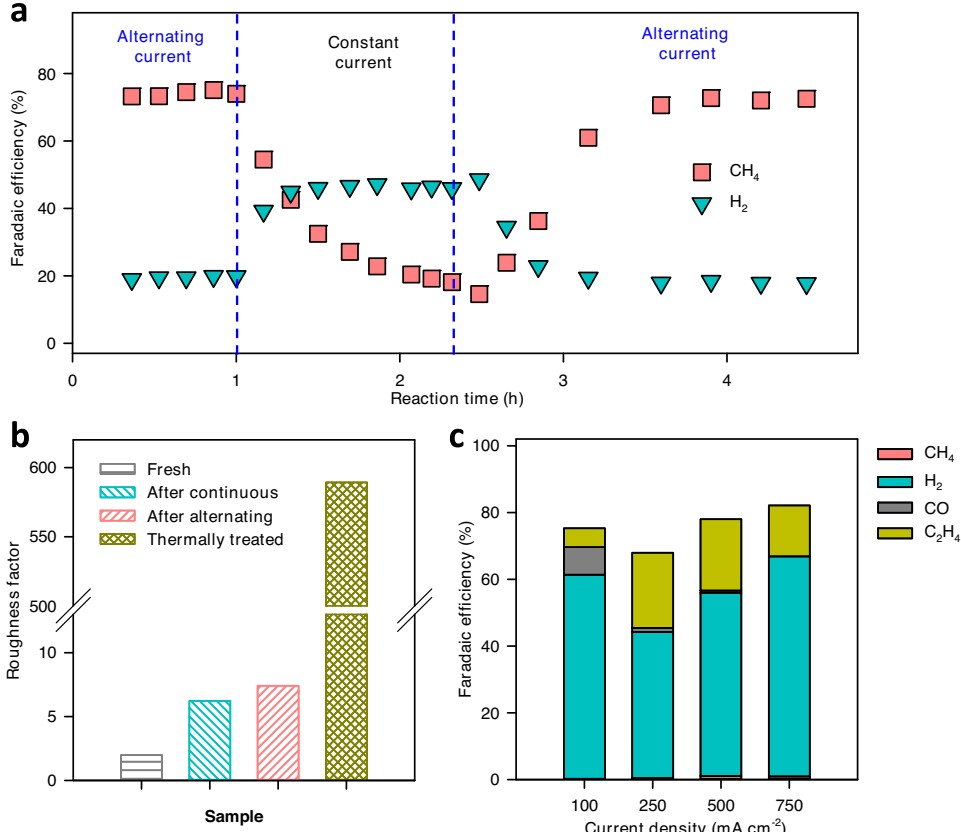

**Fig. 5 | Roles of operation mode and surface roughness factor. a** Product distribution of Cu mesh over time operated using alternating currents (oxidation current density of 2.5 mA cm$^{-2}$ and oxidation time of 5 s; reduction time of 25 s and $CO_2$ reduction current of 250 mA cm$^{-2}$), followed by constant reduction current at 250 mA cm$^{-2}$ and then alternating currents. **b** Roughness factors of fresh Cu mesh, Cu mesh after constant reduction current operation, Cu mesh after alternating current operation, and thermally treated Cu mesh after alternating current operation. **c** Product distribution of thermally treated Cu mesh using alternating current operation. Samples after reaction were collected after being tested at 100, 250, 500, and 750 mA cm$^{-2}$ current densities for 40 min at each current density (total reaction time of 160 min). For samples with alternating current operations, the oxidation current density was 2.5 mA cm$^{-2}$, the oxidation and reduction times were 5 s and 25 s, respectively.

To study the changes in the surface composition of the catalysts induced by the different operating protocols, we performed X-ray photoelectron spectroscopy (XPS) analyses of the samples before and after $CO_2$ reduction reactions (Figs. 4g, h, S15). All samples show the presence of both metallic copper (Cu), characterized by the peak at a binding energy of 932 eV, and copper oxide (CuO), indicated by the peak at a binding energy of 933.5 eV (Fig. 4g, h). The ratio of the two deconvoluted peaks at 932 and 933.5 eV are similar between the two samples, indicating a similar ratio of copper oxide and metallic copper. The presence of copper oxide on the Cu electrodes after the reaction are likely due to Cu oxidation during exposure to air.

**Origin of high CH$_4$ selectivity**

Previous studies of ECR operating at low current densities of 10–50 mA cm$^{-2}$ in aqueous systems have suggested three key factors that affect hydrocarbon selectivity, including the nature of Cu active sites[44,49,50], catalyst surface roughness or loading[51–53], and the electrolyte[54]. In our system, we found that catalyst morphology and alternating current operation are two crucial factors for high CH$_4$ production. To understand the effect of each factor, we performed a series of control experiments to decouple their roles. To investigate the contribution of operating mode, we performed ECR at a current density of 250 mA cm$^{-2}$ using alternating current until a stable CH$_4$ FE of over 70% was achieved for 1 h (Fig. 5a) and then switched the reaction to the constant current mode at the same current density. The CH$_4$ FE drops quickly to 20% after 1 h of reaction while H$_2$ FE increases significantly. When the system was switched back to alternating

operating mode again, CH$_4$ FE recovered to the initial FE of over 70%. These results confirm that even with oxidation-induced activated Cu surface, alternating current operation is needed to maintain high CH$_4$ FE in aqueous fed system at high current densities.

To study the possible effect of impurities such as Ni and Fe ions, which could leach from the Ni foam anode and be transported to the cathode during alternating current operation[55], we performed ECR using stable IrO$_x$ supported on titanium felt as the anode. This yielded similar CH$_4$ FEs compared to Ni-based anodes (Fig. S16). We also intentionally added Ni and Fe ions to the catholyte with concentrations ranging from 0.1 to 1 ppm and found that Ni and Fe impurities both show detrimental effects on CH$_4$ production (Figs. S17, S18). These results confirm that potential Ni and Fe impurities do not contribute to the high CH$_4$ FE obtained in our systems.

To study the effect of surface morphology, we estimated the surface roughness factors of the catalysts from double-layer capacitance measurements. Cu electrodes after both constant and alternating current operations show similar roughness factors of 6-8 which is ca. 3 times higher than that of fresh Cu mesh (Figs. 5b, S19). We reason that a relatively low surface roughness factor is the main reason for the high CH$_4$ FE observed in our study. This is consistent with what has been demonstrated in aqueous ECR systems using Cu mesh and Cu foils as electrodes, suggesting that a balance between surface roughness and applied potential is crucial for high CH$_4$ selectivity[53]. To further confirm the effect of surface roughness, we prepared a high surface area Cu mesh electrode using a thermal treatment process[56]. With a high surface roughness factor of 590, the electrode produced

$C_2H_4$ as the main hydrocarbon product (FE of 15–25%) while $CH_4$ production is suppressed (FE less than 1%) in current densities of 250 to 750 mA cm$^{-2}$ using alternating current operation (Figs. 5c, S20). These results suggest that both low roughness factors and alternating current operation are needed to achieve high $CH_4$ FE and production rates in our aqueous solution ECR system.

Finally, the local pH in the catalyst layer, estimated from numerical simulation for the 0.3 M KHCO$_3$ catholyte, is between 10 and 12 at current densities of 100–500 mA cm$^{-2}$ (This suggests that water proton source for $CH_4$ formation (Fig. S21). Our results are consistent with previous works showing an optimal local pH of 10.5–11 for $CH_4$ formation on a relatively flat Cu surface[53]. Experiments using lower KHCO$_3$ catholyte concentration show a higher $C_2H_4$:$CH_4$ ratio (Fig. S22), further supporting the role of pH within the catalyst layer to tune hydrocarbon selectivity. Lower KHCO$_3$ concentrations have reduced buffering effects, leading to higher local pH near the flow channel (Fig. S21), presumably outside of the optimal pH range for $CH_4$ formation.

### Experimental validation of $CO_2$ sources

As discussed above, there are two sources of $CO_2$ in the aqueous solution-fed system: $CO_2$ dissolved in the electrolyte and $CO_2$ generated in-situ from protons reacting with bicarbonate. To understand the role of each source, we performed a series of controlled experiments. First, we performed ECR using $CO_2$ saturated KHCO$_3$ electrolyte and an anion exchange membrane (AEM). In this configuration, dissolved $CO_2$ is the only $CO_2$ source because the in-situ pathway is blocked due to lack of protons. We found that dissolved $CO_2$ enables high $CH_4$ selectivity (up to 52%) at low current densities of 100 mA cm$^{-2}$ and 250 mA cm$^{-2}$ (Fig. S23). At the higher current densities of 500 mA cm$^{-2}$ and 750 mA cm$^{-2}$, $CH_4$ FE is decreased to below 40%, which is much lower than those obtained using $CO_2$-saturated KHCO$_3$ and a BPM (Fig. S23). These results suggest that dissolved $CO_2$ can serve as the main $CO_2$ source for $CH_4$ production at low current densities (below 250 mA cm$^{-2}$) but is not sufficient to maintain high $CH_4$ selectivity at higher current densities (above 500 mA cm$^{-2}$).

To explore the role of in-situ generated $CO_2$ gas, we performed the reaction using an $N_2$ saturated KHCO$_3$ electrolyte and a BPM. In this configuration, the contribution of dissolved $CO_2$ is limited to the equilibrium dissolved $CO_2$ concentration of the electrolyte (~1.5 mM vs. 33 mM in a $CO_2$-saturated case[57]). While in-situ generated $CO_2$ enables the formation of $CH_4$, its FE is relatively low. The highest recorded $CH_4$ FE was 27%, achieved at 500 mA cm$^{-2}$, corresponding to a $CH_4$ partial current density of 135 mA cm$^{-2}$ (Fig. S23), which is comparable to previous reported data using in-situ generated $CO_2$ gas from bicarbonate electrolyte[27]. These results confirm that the presence of dissolved $CO_2$ is crucial for high $CH_4$ FE and both $CO_2$ sources are required for high $CH_4$ selectivity at high current densities.

We also studied the effect of catholyte KHCO$_3$ concentration on ECR performance. KHCO$_3$ concentration can influence $CO_2$ solubility, the amount of in-situ $CO_2$ generated, and the local pH on the catalyst surface. Electrolyte with low KHCO$_3$ concentration has higher $CO_2$ solubility and yields higher local pH on the catalyst surface during ECR due to reduced buffering. Meanwhile, high KHCO$_3$ concentration increases the availability of the HCO$_3^-$ ion at the membrane surface for in-situ $CO_2$ generation. At the current density of 100 mA cm$^{-2}$, the $CH_4$ FEs with 0.1 M and 0.3 M KHCO$_3$ electrolyte are much higher than that of 1 M KHCO$_3$, suggesting the important role of high local pH at low current densities (Fig. S22). At high current densities (500–750 mA cm$^{-2}$), 0.3 M KHCO$_3$ yields a substantially higher $CH_4$ FE compared to either 0.1 M or 1 M KHCO$_3$ (Fig. S22). These results suggest that a balance between dissolved $CO_2$, in-situ generated $CO_2$, and local pH is needed to achieve high $CO_2$-to-$CH_4$ conversion. Compared to the dense electrodes used in previous studies[27], the open matrix catalyst used in this work effectively utilizes dissolved $CO_2$ as the carbon source. Therefore, highly concentrated KHCO$_3$ electrolyte is

not needed to provide enough in-situ generated $CO_2$. As a result, the open matrix electrode allows the utilization of KHCO$_3$ electrolyte with relatively low concentration, which produces favorable local pH for $CO_2$ conversion.

To validate the experimental trends and the conclusions from the multiphysics model, the model was tested against various experimental configurations, including AEM operation (Fig. S24), various sparging gases (Fig. S24), and at different electrolyte concentrations (Fig. S25). The model can replicate cell performance trends in all cases.

### Product concentration and system stability

To analyze product concentration in the outlet stream, we designed a system in which the gas product is separated from $CO_2$ bubbling solution. Gas and liquid electrolyte go through a separator. Gas products were collected for analysis while liquid electrolyte is recycled to the $CO_2$-saturating solution (Fig. 6a). $CH_4$, $H_2$, and $CO_2$ are the main components in the gas products. At 100 mA cm$^{-2}$, both $CH_4$ and $H_2$ concentrations are relatively low, while $CO_2$ makes up the bulk of the gas stream (Fig. 6b). At higher current densities, $CH_4$ and $H_2$ concentration increases, reaching a maximum concentration of 23.5% for $CH_4$ at 500 mA cm$^{-2}$. The molar ratio of $CH_4$ produced to unreacted $CO_2$ gas was 40.7%, exceeding the highest previously reported of 34%[27]. During the reaction, $CO_2$ gas is produced from the reaction of bicarbonate and protons from the BPM (Eq. 1). At the same time, $CO_2$ is consumed via the formation of $CH_4$ (Eq. 2) and the reaction with hydroxyl ions (OH$^-$) during the reaction.

$$CO_2 + OH^- \rightarrow HCO_3^- \tag{3}$$

In principle, all $CO_2$ produced from Eq. (1) can be converted to $CH_4$ or HCO$_3^-$ via Eqs. (2) and (3) because the amount of H$^+$ and OH$^-$ produced are equal. Excess amount of $CO_2$ observed in the gas stream could be due to the low efficiency of the reaction (3) or slow absorption from gas phase $CO_2$ bubbles into the bulk electrolyte. We reason that a well-designed gas/liquid separator that allows efficient $CO_2$ conversion to bicarbonate would significantly reduce the amount of $CO_2$ gas in the product stream.

To evaluate the stability of the system, we performed the reaction at current densities of 250 and 500 mA cm$^{-2}$ and tracked the gas products over time (Figs. 6c, d, S26). At 250 mA cm$^{-2}$, a $CH_4$ FE in the range of 70–75% was maintained over a period of 12 h. $H_2$ and CO FEs were relatively constant while ethylene FE increased slightly from 2% to 4% after 12 h of reaction. A similar trend was observed at the current density of 500 mA cm$^{-2}$ with $CH_4$ FE being stable at around 65 - 70% throughout the experiment of 12 h in duration (Fig. 6d).

### Discussion

High $CH_4$ selectivity at relatively high partial current density can be obtained using an alkaline flow cell system (Fig. 7a and Table S1)[11–17]. However, carbonate formation in this system requires a significant amount of energy for electrolyte regeneration. Neutral-pH electrolytes can reduce carbonate formation but often lead to lower $CH_4$ selectivity at relatively lower partial current density compared to alkaline flow cells (Fig. 7a and Table S1)[18–22]. While high $CH_4$ selectivity has been achieved with flow cells, the flow cell platform typically exhibits a large cell resistance, requiring high cell voltages to achieve high current densities. Thus, energy efficiency has not been reported for flow cell systems[11–22]. MEA cells using an anion exchange membrane can produce $CH_4$ at relatively high selectivity and energy efficiency (Fig. 7a, b)[2]. However, $CO_2$ crossover with anion exchange membranes requires additional $CO_2$ separation. In both flow cell and MEA systems, $CH_4$ produced is diluted with the unreacted $CO_2$ gas stream leading to low $CH_4$ product concentration (Fig. 7c). Previous bicarbonate-fed systems produced $CH_4$ with relatively high concentration and minimized $CO_2$ crossover. However, they exhibit low energy efficiency, FE, and partial

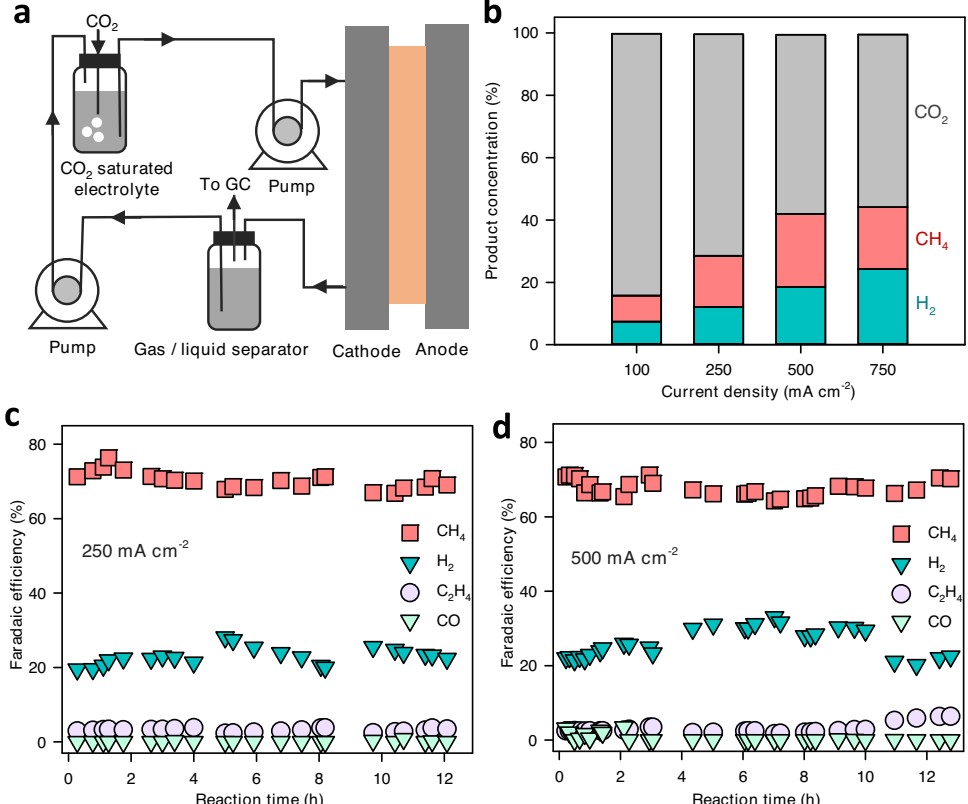

**Fig. 6 | Product concentration and system stability. a** Schematic illustration of the experimental setup for analyzing product concentration. **b** $H_2$, $CH_4$, and $CO_2$ concentrations in the outlet gas stream at different current densities. Gas product distribution over time at current densities of 250 mA cm$^{-2}$ (**c**) and 500 mA cm$^{-2}$ (**d**) shows the stability of the system. The operation conditions in Fig. (**b**)–(**d**) were as follows: 25 s reduction time, 2.5 mA cm$^{-2}$ oxidation current density, and 5 s oxidation time.

current density compared to those obtained with flow cell and MEA systems (Fig. 7)[27].

We developed a bicarbonate-fed system which achieves the high $CH_4$ selectivity (>70%) typically found in flow cell configurations using gas-phase $CO_2$ electrolysis. The obtained $CH_4$ partial current density of over 400 mA cm$^{-2}$ exceeds most of what has been previously achieved in both flow cell and MEA systems (Fig. 7a). Our system delivers an energy efficiency of 15–18% in the current range of 250–500 mA cm$^{-2}$ which are comparable to previous MEA systems while minimizing carbonate crossover (Fig. 7b). Particularly, our dissolved $CO_2$ and bicarbonate-fed system also delivers the highest $CH_4$ product concentration compared to all previous $CO_2$-to-$CH_4$ systems (Fig. 7c). The performance of the architecture is facilitated by in-situ surface activation of the copper catalyst combined with facile liquid transport enabled by an open matrix structure. An optimized surface activation strategy is presented which balances surface reconstruction with maximizing effective current density. The large pore structure of the catalyst allows both dissolved $CO_2$ and $CO_2$ generated in situ to be efficiently utilized, as confirmed by both experiment and multiphysics modeling. Further improvements to the system performance are possible through the suppression of $CO_2$ bubbling and optimization of the gas/liquid separation scheme.

## Methods

### Electrochemical $CO_2$ reduction
Electrochemical measurements were performed in a two-electrode MEA flow cell. Details of the electrolysis cell and the electrolysis system employed in this work are described in Fig. S5. In the MEA flow cell, copper mesh (100 pores per inch (ppi)) and nickel foam (1.5 mm thickness; 80–100 ppi, MTI Corp.) were used as the cathode and

anode, respectively, and were separated by a bipolar membrane (Fumasep) or an anion exchange membrane (Fumasep). The exposed sizes of both the cathode and anode electrodes were 1 cm × 2 cm (a geometric area of 2 cm$^2$ was used for all current density calculations). The copper mesh was configured to be in direct contact with the membrane. The polytetrafluoroethylene (PTFE) gasket was used to prevent contact between anode and cathode titanium flow plates. A PTFE spacer was used to avoid direct contact between the Cu mesh and the cathode flow plate.

For the electrolysis system, a potentiostat (Autolab PGSTAT204) with a current booster (Metrohm Autolab, 10 A) was used for all experiments. Two peristaltic pumps were used to circulate the anolyte (1 M KOH; 200 mL) and catholyte (0.1 M, 0.3 M, or 1 M KHCO$_3$; 1000 mL) between the reservoirs and the electrochemical cell at a flow rate of 30 mL min$^{-1}$. Before the measurement, $CO_2$ gas (Praxair, 99.99%) or N$_2$ gas (Praxair, 99.99%) was purged in the KHCO$_3$ aqueous solution for at least 30 min. $CO_2$ was continuously purged into the catholyte throughout the experimental process (during both oxidation and reduction cycles) at a flow rate of 50 standard cubic centimeters per minute (sccm). Electrolysis was carried out for 2500 s at each tested current density, with the reaction gaseous product being analyzed every 15 min. Gas products coming out from the cell were carried by the $CO_2$ or N$_2$ stream bubbled through the catholyte reservoir to a 6-way valve where it was injected to a gas chromatography (GC, PerkinElmer Clarus 590) for quantification. The GC is equipped with a thermal conductivity detector (TCD) operated at 180 °C and a flame ionization detector (FID) operated at 250 °C. A molecular sieve (5A) packed column (Supelco) connected to the TCD was used to analyze CO, $CO_2$, and H$_2$ products while a Carboxen-1000 packed column (Supelco) connected to the FID was employed to quantify $CH_4$, $C_2H_4$ and other

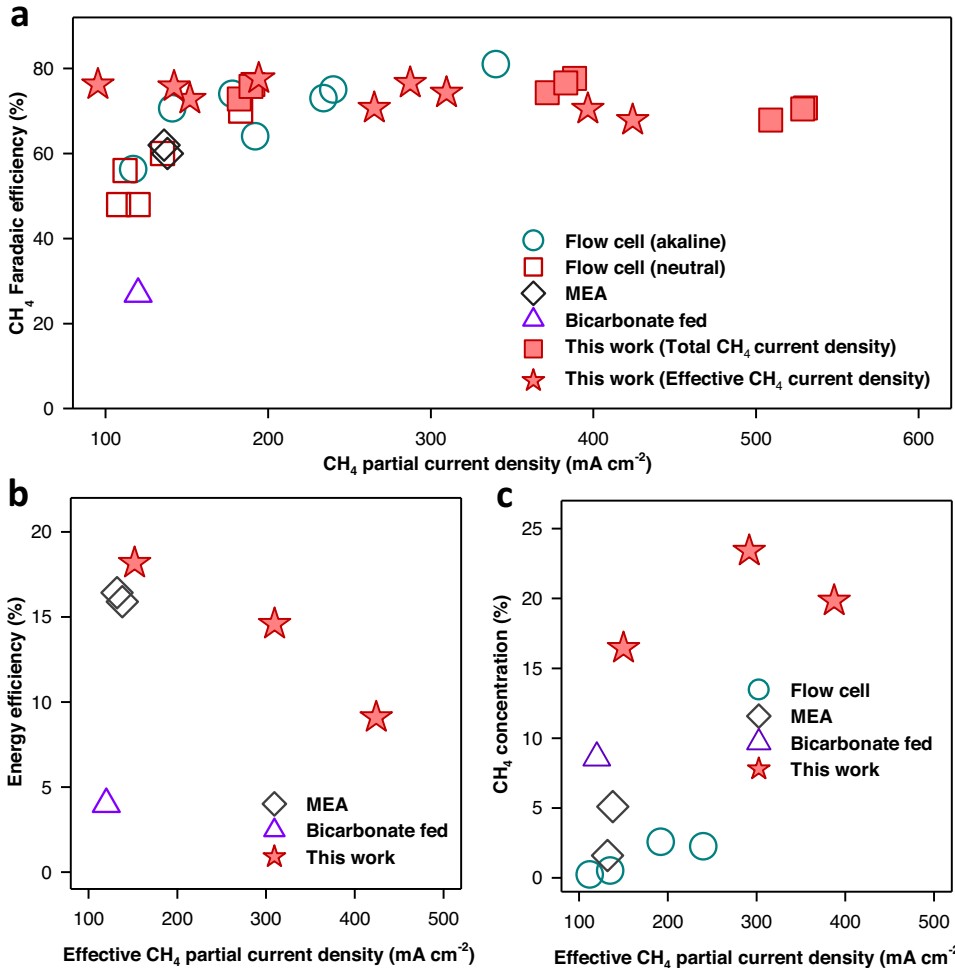

**Fig. 7 | Performance comparison between ECR systems.** Comparison of our work with previous studies on $CO_2$-to-$CH_4$ conversion with partial current densities over 100 mA cm$^{-2}$ using a flow cell (with alkaline electrolyte[11–17] and pH-neutral electrolyte[18–22]), an MEA cell with AEM[2, 11], and bicarbonate fed using BPM[27].

**a** $CH_4$ Faradaic efficiency. **b** Energy efficiency. **c** $CH_4$ product concentration at the outlet stream. Effective $CH_4$ partial current density was used for Fig. 7b, c. Detailed data are shown in Table S1.

potential hydrocarbons. Both columns were operated at a fixed temperature of 180 °C and with Ar carrier gas (flowrate of 20 sccm).

Aliquots of the liquid products were analyzed using nuclear magnetic resonance spectroscopy (NMR)[1].H NMR spectra of freshly collected liquid products were acquired on an Auto-400 ultrashield Bruker Instrument operating at denoted spectrometer frequency given in megahertz (MHz) at 25 °C in $D_2O$ using water suppression mode, with dimethyl sulfoxide as the internal standard.

The FE of ECR gas products was calculated as follows:

$$FE_k = \frac{n_k \cdot F \cdot x_k \cdot F_m}{I} \times 100\%$$

Where $FE_k$ is the FE of product $k$ , $n_k$ is the number of electrons (for $k=$ $CH_4$, $n_k$ is 8) transferred to form gaseous product $k$ , $F$ is Faraday's constant (96,485 C mol$^{-1}$), $F_m$ is the molar flow rate of the gas outlet stream in $\frac{mol}{s}$. $x_k$ is the molar fraction of the gas product k in the gas outlet stream during reduction cycle. $I$ is the total (applied) current in Amperes ($A$ ) during reduction cycle.

Because the reduction and oxidation cycles are relatively short and the $CO_2$ is constantly flowing through the system (during both reduction and oxidation cycles), gas produced during the reduction cycle is diluted by $CO_2$ flow during the oxidation cycle. Thus, the actual molar fraction of the gas product during the reduction cycle is

calculated as follows:

$$x_k = x'_k \cdot \frac{\triangle t_r + \triangle t_o}{\triangle t_r}$$

where $x'_k$ is the diluted concentration (mixing of gas production in both oxidation and reduction cycle) measured by the Gas chromatography. $\triangle t_r$ and $\triangle t_o$ are the reduction and oxidation time in each reduction–oxidation cycle.

The FE of ECR liquid products were calculated as follows:

$$FE_l = \frac{n \cdot F \cdot \triangle \delta_l}{\triangle Q} \times 100\%$$

Where $n$ is the number of electrons transferred to form liquid product $l$, $F$ is the Faraday's constant (same as above), $\Delta\delta_l$ is the accumulated number of moles of the corresponding product $l$. $\triangle Q$ is the total charge transfer during the electrolysis.

$CH_4$ partial current density was calculated in similar way as:

$$j_{CH_4} = \frac{I \cdot FE_{CH_4}}{a}$$

Where $j_{CH_4}$ is the partial current density of $CH_4$. $a$ is the geometric surface area ($a$ = 2 cm$^2$ in our system). $I$ is the total current during the reduction cycle.

Effective $CH_4$ current density was calculated as follows:

$$J_{CH_4,eff} = I \cdot FE_{CH_4} \cdot \left( \frac{\triangle t_r}{\triangle t_r + \triangle t_o} \right)$$

where $J_{CH_4,eff}$ is the effective $CH_4$ current density. The effective current density reflects the effective operation time of the electrolyzer considering both the time for $CO_2$ conversion (reduction) and the time for catalyst regeneration (oxidation). $\triangle t_r$ and $\triangle t_o$ are the reduction and oxidation times in each reduction–oxidation cycle. The factor $\frac{\triangle t_r}{\triangle t_r + \triangle t_o}$ is used in the expression to weight operation (conversion) time to the total time.

The energy efficiency, EE, was calculated as follows:

$$EE = \left( \frac{E^o \cdot FE_{CH_4}}{V_{cell}} \right)$$

Where $V_{cell}$ is the applied cell potential for a given geometric current density. $E^o$ = 1.06 is the equilibrium potential for the reaction: $CO_2 + 2H_2O = CH_4 + 2O_2$.

## $IrO_x$/Ti felt preparation

The $IrO_x$/Ti felt was prepared using a drop-casting method. A mixture of 40 mg of $IrO_x$ (Fuel Cell Store) and 160 μL of 5 wt% Nafion perfluorinated resin solution (Sigma-Aldrich) in 5 ml methanol was sonicated for 30 min. Then 1 mL of the resulting mixed solution was drop-casted onto a Ti felt substrate (1 cm × 2 cm; Fuel Cell Store). The sample was allowed to dry overnight in ambient conditions.

## High surface area Cu electrode preparation

The high surface area Cu electrode was prepared by thermal treatment. The Cu mesh was kept in the furnace and heated to 500 °C at a ramp rate of 5 °C min$^{-1}$. The furnace temperature was maintained at 500 °C for 3 h before it was cooled down to room temperature. The resulting high surface area Cu mesh was pretreated and electro-reduced at a current density of 20 mA cm$^{-2}$ for 300 s in H-cell with a 0.3 M KHCO$_3$ catholyte solution before performing an alternating reduction/oxidation reaction.

## Electrochemical double-layer capacitance measurement

Electrochemical double-layer capacitance was determined by measuring cyclic voltammetry (CV) in an H-cell, made up of two different chambers and separated by an anion exchange membrane. A three-electrode setup was used, comprised of the working electrode and a KCl saturated reference electrode (Ag/AgCl electrode) in the cathode chamber and a platinum counter electrode placed in the anode chamber. All potentials for this measurement were determined against the Ag/AgCl reference electrode. The dimension of the working electrode which was immersed in the electrolyte is 1 cm × 1 cm. First, the potential range of the non-Faradaic current regime was determined from CV. CV measurements were then conducted in a $CO_2$ saturated 0.3 M KHCO$_3$ electrolyte solution by sweeping the potential in the non-faradaic region between −0.55 and −0.65 V vs. Ag/AgCl with 3 cycles for each scan rate of 0.1, 0.2, 0.4, 0.6 0.8, 1, 1.2, and 1.4 V s$^{-1}$. For the thermally treated Cu mesh, scan rates of 0.005, 0.01, 0.02, 0.04, and 0.06 V s$^{-1}$ were used. Roughness factors of Cu electrodes were estimated by comparing their capacitances against that of an ideally smooth Cu surface (0.029 mF cm$^{-2}$)[56].

## Characterizations

Catalyst surface morphology was characterized by a ZEISS Auriga FE-SEM operated at 3 kV. XPS measurements were performed using a Thermo Scientific K-Alpha spectrophotometer with a monochromated Al Kα X-ray radiation source.

## Multiphysics modeling

A one-dimensional multiphysics model was developed to describe the catalyst layer and cation exchange membrane transport dynamics. The transport and reaction of aqueous HCO$_3^-$, K$^+$, CO$_3^{2-}$, OH$^-$, H$^+$, and CO$_2$ are considered in the model, treated with dilute solution theory. The transport of species is governed by the Nernst-Planck equation, with the convection term neglected and electroneutrality imposed. The effective diffusion coefficient is determined with the Bruggeman correction. The porosity of the open matrix catalyst is assumed to be 0.99, based on the free transport expected in a mesoscale copper mesh, and the dense matrix 0.8[38]), matching previous works, with the gaseous fraction set to 0.2 in both cases[38]. The current distribution is determined using Ohm's law. The electrochemical kinetics are described by the concentration-dependent Butler–Volmer equation[57]. Only the H$_2$ and the CH$_4$ evolution reactions are considered due to the low experimentally recorded rates of other competing reactions. The electrochemical rates are fit to the experimental performance of the copper mesh catalyst under alternating current (5 s oxidation, 25 s reduction) for $CO_2$ sparging and N$_2$ sparging cases. The model performance was tested against the $CO_2$ sparging with an AEM case (Fig. S24). We assume that the dense matrix and open matrix catalysts share the same electrochemical rates because it is not expected that macroscale porosity would affect catalyst morphology and performance[38]. The electrode-specific surface area is assumed to be 10$^4$ m$^{-1}$ for open matrix catalyst and 10$^5$ m$^{-1}$ for the dense matrix to account for reduced specific surface area when comparing a copper mesh to a copper foam. The Donnan equilibrium boundary condition was used to describe the charge discontinuity between the catalyst layer and the cation exchange layer (CEL). The CEL has a fixed space charge density of −1.75 C m$^{-3}$ ref. 58. The water hydration (mol H$_2$O per mol SO$_3^-$) in the CEL was determined to be 6[59], which defines CEL diffusivity, as in ref. 60 Finite-rate carbonate buffer kinetics were included throughout the catalyst and CEL domains to model HCO$_3^-$, CO$_3^{2-}$, OH$^-$, H$^+$, and CO$_2$ equilibrium[37]. Henry's law was used to model $CO_2$ phase transfer from liquid to vapor phase, and it was assumed that any $CO_2$ bubbled away could not re-dissolve into the solution due to the fast timescales associated with bubble advection[38,39].

The boundary condition at the end of the catalyst layer corresponds to the interface between the catalyst layer and flow plate. A mass flux boundary condition corresponding to finite mass flux from a channel with a constant Sherwood number was imposed. An assumed Sherwood number of 36.6 was used for the open matrix case and 3.66 (corresponding to laminar channel flow) for the dense matrix case to account for unsteady convection increasing catholyte flux in the open matrix case. The catholyte used was 0.1 M, 0.3 M, or 1.0 M KHCO$_3$ where indicated. $CO_2$ sparging was imposed by setting the $CO_2$ concentration to the $CO_2$ saturation concentration (33 mM), and N$_2$ sparging by setting the $CO_2$ concentration to the equilibrium concentration (1.5 mM for 0.3 M KHCO$_3$)[59]. All other species concentrations are set to their equilibrium values, shown in Table S2. The solid-phase electric potential on the flow-plate interface is set to the half-cell potential (versus SHE), varied from 0 V to −2.1 V. The boundary condition on the other side of the domain corresponds to the interface between the cation exchange layer and anion exchange layer in a BPM. To account for the proton flux from the BPM, a boundary flux of protons equal to the integrated current density divided by the Faraday constant times the transference number is imposed as in ref. 38 The net ionic current is from K$^+$ across the membrane. All other species (HCO$_3^-$, CO$_3^{2-}$, OH$^-$, and CO$_2$) have a no-flux condition at the CEL/AEL boundary. A transference number of 0.75 is used for the BPM cases (somewhat lower than in Lees et al. due to the comparatively lower catholyte K$^+$ concentration compared to the anolyte), and 0 for the

AEM case. The electrolyte potential at the CEL/AEL boundary is set to 0 V.

The model is implemented in COMSOL Multiphysics version 6.0 and solved, assuming steady state, using the PARDISO solver. The 1D domain is discretized into elements of maximum size 0.5 μm, with element sizes of 0.02 μm close to boundaries. The results were found to be independent of further mesh refinement (Fig. S27). Additional model parameters and definitions are included in Table S2 of the SI.

## Data availability

All the data supporting the findings of this study are available within the article and its Supplementary Information and Source Data file. Source data are provided in this paper.

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

## Acknowledgements

C.A.O., G.G., and J.C. acknowledge financial support from Queen's University. C.T.D. acknowledges the financial support from the Natural Sciences and Engineering Research Council of Canada (NSERC), Canada Foundation for Innovation (CFI), and Queen's University. The authors thank Peter Brodersen and Prof. Oleksandr Voznyy for XPS analyses. F.P.G.d.A. and V.G. acknowledge funding from CEX2019- 000910-S (MCIN/AEI/10.13039/501100011033), Fundació Cellex, Fundació Mir-Puig, Generalitat de Catalunya through CERCA and the La Caixa Foundation (100010434, E.U. Horizon 2020 Marie Skłodowska-Curie grant agreement 847648).

## Author contributions

C.T.D. supervised the project. C.A.O. and G.G. conducted the experiments and data processing. J.C. performed modeling. V.G. and F.P.G.A. performed SEM analysis. C.A.O., G.G., J.C., and C.T.D. wrote the manuscript. All authors discussed the results and assisted during manuscript preparation.

## Competing interests

The authors declare no competing interests.
