## [Peer review file · Nature Communications]

REVIEWER COMMENTS

Reviewer #1 (Remarks to the Author):

This manuscript describes a means for operating a membrane electrode assembly (MEA) electrochemical cell in such a way as to obtain a high total current density ($> 100\text{mA}/\text{cm}^2$), a high CH_4 FE, and relatively little CO_2 crossover from the cathode to anode portions of the MEA. These goals are achieved by feeding the cathode flow channel with CO_2 -saturated KHCO_3 , whereas a solution of KOH is fed to the anode flow channel. A Cu mesh is used as the cathode and XX is used as the anode catalyst. A bipolar membrane is used in reverse bias mode to separate the cathode from the anode. To obtain high CH_4 FEs, the authors find that it is necessary to operate the MEA in a pulsed mode, so that the Cu catalyst can undergo short (5 s) periods under oxidizing conditions and then longer periods ($\sim 25\text{s}$) under reducing conditions. The empirical results in this manuscript are interesting in that they show how a combination of variables must be manipulated in order to achieve high CH_4 FE and current density. What is unsatisfactory, though, are the interpretations of many of the phenomena reported, which are done largely through qualitative argumentation, rather than simulation based on a physical model of the processes observed. Illustrations of the weaknesses in the interpretation of experimental results are presented below.

1. Fig. 1 presents the results of a one-dimensional model of a bicarbonate-fed electrolyzer with the purpose of demonstrating why an open Cu matrix is superior to a dense matrix. Unfortunately, neither Fig. 1a or the text give a clear picture of what is being modeled in the cathode chamber. For example, why is the catalyst shown to be orthogonal to the gas flow channel and the CEL portion of the bipolar membrane. The physics of the model are not explained nor are the boundary conditions used at the CEL-AEL interface and at the gas flow channel clearly described. Likewise, it is not clear how the Tafel parameters for the Cu catalyst were obtained and why they are assumed (presumably) to be the same for the two forms of catalyst. As a consequence, of these omissions, it is not possible to understand exactly why the open Cu matrix is superior to the dense one. Either the authors need to give much more detail or, alternatively, dispense with the simulations and just present experimental data to support their conclusion that an open Cu matrix produces more CH_4 .
2. Fig. 2a gives a clear representation of the MEA system used for the experimental portion of the work. However, Figs. 2b-d are hard to comprehend. What are shown are cartoons of current flow vs reaction time but no units are given on either axis of the plots. It is also not clear how the authors varied the current flow in the cell from oxidizing to reducing. Was this done by changing the magnitude and the polarity of the cathode relative to the anode. If so, the authors need to show the magnitudes of these voltages and currents.
3. Fig. 4 shows SEM images of the cathode before and after subsection to pulsing. What is missing from this figure is a low magnification image of the Cu mesh, so that one can understand the size of the wires comprising the mesh. The roughening of the Cu surface via repeated oxidation and reduction is well known from the work of Roldan Cuenya and many other authors. Since the roughening is done with the mesh in the MEA, what is not reported by the authors is what happens to the Ni foam used as the anode. Under normal OER conditions Ni in alkaline solution (e.g., KOH) transforms to NiOOH (e.g., *J. Phys. Chem. C*, 119, 7243-7254 (2015); *J. Phys. Chem. C*, 119, 18303-18316 (2015); *J. Phys. D. Appl. Phys.*, 55, 323003 (2022); and references cited in these papers). However, this material is a poor OER catalyst, but it rapidly improves its performance as very small amounts of Fe, present as impurities in commercial KOH , are incorporated into the NiOOH lattice to form $\text{Fe}_x\text{Ni}_{(1-x)}\text{OOH}$. Therefore, the authors need to characterize their anode to assess the amount of Fe present. Since it is known that overtime Fe cations will leach from the $\text{Fe}_x\text{Ni}_{(1-x)}\text{OOH}$ anode and transport to the cathode, the authors

need to assess the presence of Fe on the Cu cathode. This can only be done by Inelastic Ion Scattering (ISS), since XPS is not sufficiently surface sensitive. This is an important issue because Fe contamination of Cu may be the real cause of the enhanced FE for CH₄, rather than reconstruction of the Cu surface. The authors should also note that both experimental and theoretical work reported in (e.g., Chem. Mater., 32, 3304-3311 (2020); ACS Energy Lett., 5, 1206-1214 (2020)) show that roughened CuO does not produce CH₄ but, rather, is highly selective to C₂H₄. All of this said, the authors should try to rerun their experiments using IrO_x as the OER catalyst, since it is stable in alkaline media. 4. The results presented in Figs. 5 and 6 are nice because they demonstrate how the MEA system can be tuned to produce a high partial current of CH₄ over an extended period of time. What is missing though from the text is a clear explanation of why the system studied is selective to CH₄ vs C₂H₄, as one might expect. In summary, the authors have presented some intriguing results; however, the interpretation of these results is currently incomplete and does not meet the standards expected for papers published in Nature Communications. A major revision of this work is needed for it to become acceptable.

Reviewer #2 (Remarks to the Author):

The authors have investigated and developed a liquid-fed electrolytic system to convert CO₂ from an aqueous solution to CH₄. In the electrolysis, they employed mainly CO₂-saturated KHCO₃ aqueous solution and coupled the solution with an open matrix Cu mesh catalyst to enhance the supply of dissolved CO₂ molecules as the main CO₂ source to the cathode reaction site. The periodic oxidation treatment forms and keeps active porous Cu surface, which the authors think is important to keep the high Faradaic efficiency for CH₄ production (FE(CH₄)) of over 70% in a wide current density range. The aforementioned results and the authors' strategy to improve FE(CH₄) would be really interesting and significant in the research fields of both gas- and liquid-fed CO₂ electrolysis.

The present work, however, should have similar works previously reported by other research groups, which were conducted under similar experimental conditions. It would be necessary for the authors to clearly describe and discuss the difference between their present work and previous similar works and explain key factors to the good performance they achieved with quantitative scientific evidences. The reviewer requests the authors to respond to the following comments. (Reasons why the reviewer made these comments are represented in the attached reviewer report.)

[#1] Please describe the details of the multiphysics modeling performed in this work (all of the parameters and their values used in the modeling), which are necessary for readers to fully understand the situation and reaction environment the authors consider and to compare the simulation results.

[#2] The following description seen in the "Modeling the reaction environment and CO₂ availability" Subsection (p. 4) should be revised to avoid misunderstanding, for example, by clearly explaining that the reaction environment considered in the modeling and in the electrolysis of this work is different from that considered in previous works like Ref. 27; "State-of-the-art bicarbonate electrolyzers rely on catalysts within a dense porous matrix with the CO₂ source in-situ generation from bicarbonate. These electrolyzers often suffer from low CH₄ FE at industrially relevant current densities, as predicted in our simulations (Figure 1b)".

[#3] Please examine the validity of the simulation results by comparing them with corresponding experimental results.

[#4] Please clearly express the details of the electrolysis cell and the electrolysis system employed in this work.

[#5] Please show and compare SEM images of different domains at the surface of Cu mesh whose appearances are quite different before and after the CO₂ electrolysis. Also, the authors should quantitatively express the surface states (especially the porous feature) of the Cu mesh before and after the CO₂ electrolysis, for example, by using surface area and surface roughness.

[#6] Please verify and show the reproducibility of the phenomenon that Cu nanoparticles are evolved under a constant-current condition and also explain driving forces to evolve Cu nanoparticles on the surface in a reducing environment.

[#7] Please clearly describe standard electrolysis conditions in this work (e.g., the concentration of K⁺ ion in the catholyte), electrolysis duration for one current density condition, and how to conduct online gas chromatography (GC) analysis (e.g., the method to introduce gaseous products in the electrolysis to the GC instrument, the flow rate of carrier gas if used).

[#8] Please clearly describe the definition of the total current considered in this work. The value of the total current is important and affects Faradaic efficiency and energy efficiency reported in this work. The definition of Faradaic efficiency used in this work should be reviewed on the basis of the following

comment. Not only the reduction current but also the oxidation current contributes to total electricity consumption. Thus, Faradaic efficiency (or coulombic efficiency) should be the molar ratio of produced CH₄ multiplied by the number of electron consumed (= 8) to electrons flowing during the entire reduction-oxidation operation. From this point of view, the reviewer feels that it would be easy to estimate the quantity of electricity during the reduction-oxidation operation while it would not be simple to estimate (weighted average) total current.

[Additional comment] In this work, not only CO₂ reduction and hydrogen evolution reaction but also hydrogen oxidation reaction and oxygen evolution reaction during oxidation and oxygen reduction reaction during reduction could occur in the alternating reduction-oxidation operation. The absolute value of the oxidation current is much smaller than that of the reduction current, and the period of the oxidation is shorter than that of the reduction. Hence, the contribution of hydrogen oxidation (consumption) to decreasing FE(H₂) might be small. However, the reviewer recommends the authors to mention the possibilities that redox reactions of byproducts (hydrogen and oxygen) might occur.

Reviewer report

High-Rate and Selective Conversion of CO₂ from Aqueous Solutions to Hydrocarbons

Cornelius A. Obasanjo, Guorui Gao, Jackson Crane, Viktoria Golovanova,
F. Pelayo García de Arquer, and Cao-Thang Dinh

Key results

The authors have investigated and developed a liquid-fed electrolytic system to convert CO₂ from an aqueous solution to CH₄. In the electrolysis, they employed mainly CO₂-saturated KHCO₃ aqueous solution and coupled the solution with an open matrix Cu mesh catalyst to enhance the supply of dissolved CO₂ molecules as the main CO₂ source to the cathode reaction site. The periodic oxidation treatment forms and keeps active porous Cu surface, which the authors think is important to keep the high Faradaic efficiency for CH₄ production (FE(CH₄)) of over 70% in a wide current density range. The aforementioned results and the authors' strategy to improve FE(CH₄) would be really interesting and significant in the research fields of both gas- and liquid-fed CO₂ electrolysis.

Significance / Comparison with relevant previous works

The present work, however, should have similar works previously reported by other research groups, which were conducted under similar experimental conditions. It would be necessary for the authors to clearly describe and discuss the difference between their present work and previous similar works and explain key factors to the good performance they achieved with quantitative scientific evidences. Some instances of previous similar works are listed below.

[A] (Bi)carbonate solution-fed electrolysis (in which HCO₃⁻ ions are the main CO₂ source)

(Ref. 27) E. W. Lees, A. Liu, J. C. Bui, S. Ren, A. Z. Weber, and C. P. Berlinguette, "Electrolytic methane production from reactive carbon solutions", *ACS Energy Lett.*, **7**, 1712 (2022). — *Electrochemical reduction of CO₂ in situ generated in the vicinity of the cathode reaction site to form CH₄*

[B] Liquid-fed electrolysis using CO₂-saturated KHCO₃ aqueous solution (in which neutral CO₂ molecules are the main CO₂ source)

(Ref. 43) J. Timoshenko *et al.*, "Steering the structure and selectivity of CO₂ electroreduction catalysts by potential pulses", *Nat. Catal.*, **5**, 259 (2022). — *Periodic reduction–oxidation operation to tune the surface state of Cu catalyst and also the product distribution*

Y. Cai *et al.*, "Insights on forming N,O-coordinated Cu single-atom catalysts for

electrochemical reduction CO₂ to methane”, *Nat. Commun.*, 12:586 (2021). —*Electrochemical reduction of CO₂ in CO₂-saturated KHCO₃ aqueous solution to form CH₄ with FE(CH₄) > 70%*

S. Wei *et al.*, “Construction of single-atom copper sites with low coordination number for efficient CO₂ electroreduction to CH₄”, *J. Mater. Chem. A*, **10**, 6187 (2022). —*Electrochemical reduction of CO₂ in CO₂-saturated KHCO₃ aqueous solution to form CH₄ with FE(CH₄) > 70%*

When comparing the present work with the above previous works, it is important to consider the following five experimental factors. Here, comparison focuses on Ref. 27 because the article is mentioned and compared in the Introduction Section of the manuscript.

[1] Structure of Cu catalyst: open matrix Cu mesh with a porous Cu surface (this work) or Cu foam (Ref. 27)

[2] Operation: electrolysis using alternating negative and positive currents (this work) or electrolysis at constant currents (Ref. 27)

[3] Gas supplied to the electrolyte solution containing bicarbonate ions (standard condition): 100% CO₂ (this work) or N₂ (Ref. 27)

[4] K⁺ concentration in the electrolyte solution: 0.1, 0.3, or 1 M (this work) or 3 M (Ref. 27)

[5] Electrolysis cell and system determining the reaction environment (i.e., the concentration and distribution of reactants and products around the cathode reaction site)

The significance and progress of this work against Ref. 27 would be the improvement of FE(CH₄) as expressed in the Introduction Section of the manuscript: for example, about 60% in this work versus about 30% in Ref. 27 at a current density of about 100 mA/cm². Based on the authors’ comments described in the Introduction Section of the manuscript (pp. 2–3), the aforementioned progress should be made by overcoming the problems of “a yet uncontrolled reaction environment” and “the lack of selective catalysts, possibly because Cu reconstruction at high current density typically favors the hydrogen evolution reaction”. The former problem of “a yet uncontrolled reaction environment” is mainly related to the factors [3]–[5] while the latter problem of “the lack of selective catalysts” is mainly related to the factors [1] and [2].

For the former problem of “a yet uncontrolled reaction environment”, the dynamic reaction environment (e.g., time-dependent concentrations and distributions of reactants and products in the vicinity of the cathode reaction site during the electrolysis) significantly affects the product distribution in the electrolysis. Catholytes used in this work and in Ref. 27 are different in terms of the concentrations of K⁺, OH[−], and dissolved CO₂. Different electrolyte solutions introduced to the cathode compartment certainly result in different reaction environment. Hence, using one-dimensional multiphysics modeling would be a good practice to quantify the effect of the dynamic reaction environment on the electrolytic performance

and/or to rationalize the progress made in this work against Ref. 27. Based on the modeling conditions, the multiphysics modeling performed in this work would consider the dynamic reaction environment appearing during the electrolysis performed in this work. Actually, gaseous CO₂ mole fraction in this work is an order of magnitude greater than that in Ref. 27. However, the bulk concentration of bicarbonate ion in the liquid phase in this work is not described in Table S2. **Here, the reviewer requests the authors to describe the details of the multiphysics modeling performed in this work (all of the parameters and their values used in the modeling), which are necessary for readers to fully understand the situation and reaction environment the authors consider and to compare the simulation results (Comment #1).** Anyway, the above modeling conditions should correspond to the reaction environment in which neutral CO₂ molecules dissolved in the electrolyte solution are supplied as the main CO₂ source to the cathode reaction site. Through the multiphysics modeling, the authors would derive a conclusion that an open matrix mesh catalyst is appropriate for liquid-fed electrolysis compared to a dense matrix porous catalyst previously reported. It is quite natural that an open matrix mesh catalyst exhibits better performance than a dense porous matrix in such a reaction environment as employed in this work, because dissolved CO₂ should be the main CO₂ source and a different reaction environment favors a different catalyst structure. In the “Modeling the reaction environment and CO₂ availability” Subsection (p. 4), however, the authors describe that “State-of-the-art bicarbonate electrolyzers rely on catalysts within a dense porous matrix with the CO₂ source in-situ generation from bicarbonate. These electrolyzers often suffer from low CH₄ FE at industrially relevant current densities, as predicted in our simulations (Figure 1b)”, which the reviewer is afraid might cause misunderstanding. **The above description seen in p. 4 should be revised to avoid misunderstanding, for example, by clearly explaining that the reaction environment considered in the modeling and in the electrolysis of this work is different from that considered in previous works like Ref. 27 (Comment #2).** In addition, **the reviewer requests the authors to examine the validity of the simulation results by comparing them with corresponding experimental results (Comment #3).** In this work, the electrolysis using a dense porous matrix Cu foam catalyst as used in Ref. 27 has not been conducted. The experimental condition employed for the constant-current electrolysis (as shown in Figure 2d) should correspond to the reaction environment considered in the modeling. However, the experimental result (FE(CH₄) ≈ 0%, Figure 2d) is not in agreement with the simulation result (FE(CH₄) ≈ 55%, as shown in Figure 1b). Meanwhile, the former problem of “a yet uncontrolled reaction environment” (pp. 2–3 in the manuscript) is also related to the factor [5]: electrolysis cell and system determining the reaction environment in a practical operation (i.e., the concentration and distribution of reactants and products around the cathode reaction site). In a liquid-fed CO₂ electrolytic system, CO₂ sources (dissolved CO₂, HCO₃⁻, and CO₃²⁻), H₂O, H⁺ ions, and electrons are considered as reactants which enter the cathode compartment. The supply rate of H⁺ ion and electrons can be understood from current (density). Water molecules

abundantly exist in the cathode compartment, and thus we would not need to consider the supply rate of H₂O. For CO₂, however, the supply rate of dissolved CO₂ molecules transported within the inlet electrolyte solution depends on the flow rate of the electrolyte solution, and the supply rate of the CO₂ molecules in situ generated from bicarbonate ions to the cathode reaction site should be influenced by the supply rate of H⁺ ions and the frequency of collisions between H⁺ and (OH⁻ and HCO₃⁻). The collision frequency should depend on the structure of the cathode flow plate as well as that of the cathode catalyst (Cu mesh). In general, the diffusion of a CO₂ molecule in the electrolyte solution is not so fast, which causes the variation in the concentration of CO₂ molecules along the direction of the liquid flow. In this sense, the structure of the cathode flow plate (i.e., a liquid flow field design, channel width, channel depth, and channel length) as well as that of the cathode catalyst also affects the position-dependent supply rate of CO₂ molecules along the direction of the liquid flow. As mentioned above, the flow rate of the electrolyte solution and the structure of the cathode flow plate are very important factors which strongly influence the reaction environment and also the electrolysis performance. However, these factors are not clearly expressed in the manuscript. **The reviewer requests the authors to clearly express the details of the electrolysis cell and the electrolysis system employed in this work (Comment #4).**

For the latter problem of “the lack of selective catalysts”, this work tackles it by combining a Cu mesh catalyst with a periodic oxidation treatment technique as investigated in Ref. 43. This combination would be a feature of the present work, which could allow the long-term operation with high FE(CH₄) in a wide current density range. As seen in the “Formation of an active Cu surface induced by oxidation” Subsection (p. 10), the authors express the selective catalyst in this work as follows; “Alternating reduction-oxidation currents are found to induce the formation of a porous Cu surface (Figure 2b, d and Figure S9). We hypothesize that the difference in surface morphology is the primary cause for the striking difference in product distribution between using fixed reduction current and alternating reduction-oxidation currents.” Namely, the alternating reduction-oxidation currents is regarded as important to form and keep the active porous Cu surface for highly selective CH₄ production. For SEM images of the Cu mesh catalyst, Figures S9a and S9b correspond to fresh Cu mesh (before the electrolysis), Figures S9c, S9d, 4b, and 4d correspond to Cu mesh after CO₂ reduction using alternating reduction-oxidation currents, and Figures 4a and 4c correspond to Cu mesh after CO₂ reduction using fixed currents. Figures 4b, 4d, S9b, and S9d are extracted from the manuscript, and the sizes of the images are adjusted to facilitate comparison of the images, as shown below (**Figures A and B** in this report). It is true that Figure S9d looks similar to Figure 4b in **Figure A** because of the same situation as after CO₂ reduction using alternating reduction-oxidation currents. However, Figure 4d in **Figure B** (corresponding to after CO₂ reduction) looks different from Figures S9d and 4b in **Figure A** (corresponding to after CO₂ reduction) but looks similar to Figure S9b in **Figure B** (corresponding to before CO₂ reduction). Based on these experimental facts mentioned above, the Cu mesh used in this work should consist of

Figure A. SEM images of Cu mesh catalyst. Both Figures S9d and 4b correspond to Cu mesh after CO₂ reduction using alternating reduction-oxidation currents.

Figure B. SEM images of Cu mesh catalyst. Figure 4d correspond to Cu mesh after CO₂ reduction using alternating reduction-oxidation currents, that is, the same situation as the above Figure A. On the other hand, Figure S9b corresponds to fresh Cu mesh (namely, before CO₂ reduction).

multiple domains whose appearances are quite different from each other. This seems to be consistent with the appearance of the fresh Cu mesh as seen in Figure S9a in the Supplementary Information. Hence, it would be quite hard to conclude only from the aforementioned SEM observation that alternating reduction-oxidation currents induce the formation of a porous Cu surface, as described in the manuscript (p. 10). **The reviewer requests the authors to show and compare SEM images of different domains at the surface of Cu mesh whose appearances are quite different before and after the CO₂ electrolysis. Also, the authors should quantitatively express the surface states (especially the porous feature) of the Cu mesh before and after the CO₂ electrolysis, for example, by using surface area and surface roughness (Comment #5).** Besides, Cu mesh after usual constant-current CO₂ electrolysis shows lots of granular structures as seen in Figures 4a and 4c in the manuscript. This is not usually observed in previous works although it is true that the reaction environment in this work might affect the surface state of the Cu mesh. **If so, the authors should verify and show the reproducibility of this phenomenon that Cu nanoparticles are evolved under a constant-current condition and also should explain driving forces to evolve Cu nanoparticles on the surface in a reducing environment (Comment #6).**

Other requests

- Detailed explanation of experimental and analytical conditions is necessary for researchers working in the relevant research fields to develop their own science and technologies based on the present work. However, the reviewer is afraid that the explanation of these conditions in the manuscript is not sufficient. **The reviewer requests the authors to clearly describe standard electrolysis conditions in this work (e.g., the concentration of K⁺ ion in the catholyte), electrolysis duration for one current density condition, and how to conduct online gas chromatography (GC) analysis (e.g., the method to introduce gaseous products in the electrolysis to the GC instrument, the flow rate of carrier gas if used)(Comment #7).**
- **Definition of the total current considered in this work should be expressed in the manuscript because the value of the total current affects Faradaic efficiency and energy efficiency. The definition of Faradaic efficiency used in this work should be reviewed on the basis of the following comment (Comment #8).** Not only the reduction current but also the oxidation current contributes to total electricity consumption. Thus, Faradaic efficiency (or coulombic efficiency) should be the molar ratio of produced CH₄ multiplied by the number of electron consumed (= 8) to electrons flowing during the entire reduction–oxidation operation. From this point of view, the reviewer feels that it would be easy to estimate the quantity of electricity during the reduction–oxidation operation while it would not be simple to estimate (weighted average) total current. Also, the reviewer cannot understand the physical meaning of the factor $\Delta t_r/(\Delta t_r + \Delta t_o)$ appearing in the

definitions of Faradaic efficiency and effective CH_4 current density. Hence, the reviewer wanted to know the definition of the total current considered in this work.

- In this work, not only CO_2 reduction and hydrogen evolution reaction but also hydrogen oxidation reaction and oxygen evolution reaction during oxidation and oxygen reduction reaction during reduction could occur in the alternating reduction-oxidation operation. The absolute value of the oxidation current is much smaller than that of the reduction current, and the period of the oxidation is shorter than that of the reduction. Hence, the contribution of hydrogen oxidation (consumption) to decreasing $\text{FE}(\text{H}_2)$ might be small. However, the reviewer recommends the authors to mention the possibilities that redox reactions of byproducts (hydrogen and oxygen) might occur.

High-Rate and Selective Conversion of CO₂ from Aqueous Solutions to Hydrocarbons

Cornelius A. Obasanjo, Guorui Gao, Jackson Crane, Viktoria Golovanova,
F. Pelayo García de Arquer, and Cao-Thang Dinh

Response to the reviewers

Reviewer #1

This manuscript describes a means for operating a membrane electrode assembly (MEA) electrochemical cell in such a way as to obtain a high total current density ($> 100\text{mA}/\text{cm}^2$), a high CH₄ FE, and relatively little CO₂ crossover from the cathode to anode portions of the MEA. These goals are achieved by feeding the cathode flow channel with CO₂-saturated KHCO₃, whereas a solution of KOH is fed to the anode flow channel. A Cu mesh is used as the cathode and XX is used as the anode catalyst. A bipolar membrane is used in reverse bias mode to separate the cathode from the anode. To obtain high CH₄ FEs, the authors find that it is necessary to operate the MEA in a pulsed mode, so that the Cu catalyst can undergo short (5 s) periods under oxidizing conditions and then longer periods (~25s) under reducing conditions.

The empirical results in this manuscript are interesting in that they show how a combination of variables must be manipulated in order to achieve high CH₄ FE and current density. What is unsatisfactory, though, are the interpretations of many of the phenomena reported, which are done largely through qualitative argumentation, rather than simulation based on a physical model of the processes observed. Illustrations of the weaknesses in the interpretation of experimental results are presented below.

We thank the reviewer for the supportive comment, as well as other constructive comments below that helped us further improve the manuscript.

Specific comments:

[#1] Fig. 1 presents the results of a one-dimensional model of a bicarbonate-fed electrolyzer with the purpose of demonstrating why an open Cu matrix is superior to a dense matrix. Unfortunately, neither Fig. 1a or the text give a clear picture of what is being modeled in the cathode chamber. For example, why is the catalyst shown to be orthogonal to the gas flow channel and the CEL portion of the bipolar membrane. The physics of the model are not explained nor are the boundary conditions used at the CEL-AEL interface and at the gas flow channel clearly described. Likewise, it is not clear how the Tafel parameters for the Cu catalyst were obtained and why they are assumed (presumably) to be the same for the two forms of catalyst. As a consequence, of these omissions, it is not possible to understand exactly why the open Cu matrix is superior to the dense one. Either the authors need to give much more detail or, alternatively, dispense with the simulations and just present experimental data to support their conclusion that an open Cu matrix produces more CH₄.

Figure 1 has been updated to clarify the modelled domain, including showing the porous catalyst structure and describing all physics considered and relevant boundary conditions. In addition, the following additional details have been added to the ‘Modelling the reaction environment and CO₂ availability’ sub-section:

“The cation exchange layer and porous copper catalyst sub-systems are considered in the model (Figure 1a and 1b). Key physics are considered including species and charge transport, electrocatalytic reactions, CO₂ phase transfer, and buffer equilibrium reactions, with constants derived from past work.³⁵⁻³⁹ A proton flux from water dissociation within the BPM, proportional to the total integrated current density, is imposed on the anion exchange layer/cation exchange layer interface. A mass flux boundary condition is imposed on the catalyst layer/flow channel interface from the bulk electrolyte.”

Revised Figure 1. Modelling reaction environment and CO_2 availability. (a) Schematic illustration of the aqueous solution fed ECR system using porous Cu cathode, Ni foam anode and bipolar membrane (BPM). (b) Schematic of modeled domain with key physics annotated (c) CO_2 flux components at three current densities, 250, 500, and 750 mA/cm^2 for dense matrix and open matrix catalysts with N_2 sparging or CO_2 sparging. 0.3 M KHCO_3 was used as electrolyte for all modelling. (d) Modeled CH_4 FE as a function of current density for a dense matrix catalyst (solid lines) and open matrix catalyst (dashed lines) with CO_2 sparging and N_2 sparging.

Substantially more detail was also added into the ‘Multiphysics modeling’ sub-section of the ‘Methods’ section, including additional clarification of the boundary conditions and the Tafel kinetics:

*“The transport and reaction of aqueous HCO_3^- , K^+ , CO_3^{2-} , OH^- , H^+ , and CO_2 are considered in the model, treated with dilute solution theory. The transport of species is governed by the Nernst-Planck equation, with the convection term neglected and electroneutrality imposed. The effective diffusion coefficient is determined with the Bruggeman correction. The porosity of the open matrix catalyst is assumed to be 0.99, based on the free transport expected in a mesoscale copper mesh, and the dense matrix 0.8 (ref.³⁸), matching previous works, with the gaseous fraction set to 0.2 in both cases³⁸. The current distribution is determined using Ohm’s law. The electrochemical kinetics are described by the concentration dependent Butler-Volmer equation⁵⁷. Only the H_2 and the CH_4 evolution reactions are considered due to the low experimentally recorded rates of other competing reactions. The electrochemical rates are fit to the experimental performance of the copper mesh catalyst under alternating current (5 s oxidation, 25 s reduction) for CO_2 sparging and N_2 sparging cases. The model performance was tested against the CO_2 sparging with an AEM case (**Figure S23**). We assume that the dense matrix and open matrix catalysts share the same electrochemical rates, because it is not expected that macroscale porosity would affect catalyst morphology and performance.³⁸ The electrode specific surface area is assumed to be 10^4 m^{-1} for open matrix catalyst and 10^5 m^{-1} for the dense matrix to account for reduced specific surface area when comparing a copper mesh to a copper foam.”*

And:

“All other species (HCO_3^- , CO_3^{2-} , OH^- , and CO_2) have a no-flux condition at the CEL/AEL boundary. A transference number of 0.75 is used for the BPM cases (somewhat lower than in Lees et al. due to the comparatively lower catholyte K^+ concentration compared to the anolyte), and 0 for the AEM case. The electrolyte potential at the CEL/AEL boundary is set to 0 V.”

Additional figures (**S23** and **S24**) in the supplemental materials are included which compare experimental and modeled cell performance, now discussed in the ‘Experimental validation of CO_2 sources’ section:

*“To validate the experimental trends and the conclusions from the multiphysics model, the model was tested against various experimental configurations, including AEM operation (**Figure S23**),*

various sparging gases (Figure S23), and at different electrolyte concentrations (Figure S24). The model can replicate cell performance trends in all cases.”

Finally, additional line items are included detailing the bulk electrolyte concentrations used for the flow channel interface in **Table S2** of the Supplementary Information.

[#2] Fig. 2a gives a clear representation of the MEA system used for the experimental portion of the work. However, Figs. 2b-d are hard to comprehend. What are shown are cartoons of current flow vs reaction time, but no units are given on either axis of the plots. It is also not clear how the authors varied the current flow in the cell from oxidizing to reducing. Was this done by changing the magnitude and the polarity of the cathode relative to the anode. If so, the authors need to show the magnitudes of these voltages and currents.

We have revised the Figure 2 as follows:

Figure 2a has been moved to Figure 1 in the revised version to discuss the cell configuration in the modelling part. Figure 2b has been moved to the supporting information together with a photograph of the Cu mesh (New Figure S4) to show the size of Cu wire and pore structure of the mesh. The actual values of current versus time for the three operating conditions, including constant current, cyclic voltammetry plus constant current, and alternating current, have been included in the Revised Figure 2 to show the magnitudes of the oxidation and reduction currents.

We applied a square-wave alternating current approach in which the electrical current is alternating between a negative and a positive current. The transition from oxidation and reduction currents occurs nearly instantly. A new figure highlighting the actual values of current during 90 s reaction using the alternating current approach has been added (New Figure 2d). We have added these details to the revised manuscript:

“To this end, we applied a square-wave alternating current in which the electrical current alternated between a negative and a positive current. Using this operation mode, the catalyst is periodically oxidized during CO₂ reduction (Figure 2c and 2d).”

Revised Figure 2. Electrocatalytic CO₂ reduction with constant and alternating current operations. (a) Product distribution of Cu mesh over time operated at a constant reduction current density of 250 mA/cm². (b) Product distribution of Cu mesh after catalyst surface activation using cyclic voltammetry (CV) between -20 and 10 mA/cm² for 10 cycles, followed by operation at a constant reduction current density of 250 mA/cm². (c) Product distribution of Cu mesh operated using alternating negative and positive currents (oxidation current density of 0.5 mA/cm² and oxidation time of 3 s; reduction time of 25 s and CO₂ reduction current of 250 mA/cm²). (d) Magnification in time of the square-wave alternating reduction and oxidation current density in (c).

[#3] Fig. 4 shows SEM images of the cathode before and after subjection to pulsing. What is missing from this figure is a low magnification image of the Cu mesh, so that one can understand

the size of the wires comprising the mesh. The roughening of the Cu surface via repeated oxidation and reduction is well known from the work of Roldan Cuenya and many other authors. Since the roughening is done with the mesh in the MEA, what is not reported by the authors is what happens to the Ni foam used as the anode. Under normal OER conditions Ni in alkaline solution (e.g., KOH) transforms to NiOOH (e.g., J. Phys. Chem. C, 119, 7243-7254 (2015); J. Phys. Chem. C, 119, 18303-18316 (2015); J. Phys. D. Appl. Phys., 55, 323003 (2022); and references cited in these papers). However, this material is a poor OER catalyst, but it rapidly improves its performance as very small amounts of Fe, present as impurities in commercial KOH, are incorporated into the NiOOH lattice to form $\text{Fe}_x\text{Ni}_{(1-x)}\text{OOH}$. Therefore, the authors need to characterize their anode to assess the amount of Fe present. Since it is known that overtime Fe cations will leach from the $\text{Fe}_x\text{Ni}_{(1-x)}\text{OOH}$ anode and transport to the cathode, the authors need to assess the presence of Fe on the Cu cathode. This can only be done by Inelastic Ion Scattering (ISS), since XPS is not sufficiently surface sensitive. This is an important issue because Fe contamination of Cu may be the real cause of the enhanced FE for CH_4 , rather than reconstruction of the Cu surface. The authors should also note that both experimental and theoretical work reported in (e.g., Chem. Matls, 32, 3304-3311 (2020); ACS Energ. Letts. 5, 1206-1214 (2020)) show that roughened Cu^0 does not produce CH_4 but, rather is highly selective to C_2H_4 . All of this said, the authors should try to rerun their experiments using IrO_x as the OER catalyst, since it is stable in alkaline media.

We thank the reviewer for the suggestion related to Cu mesh size and insights related to the potential effect of Ni and Fe impurities. We also appreciate that the reviewer points out the effect of roughness factor on CH_4 and C_2H_4 selectivity. We have addressed these points as follows:

Size of the pore and wire of Cu mesh:

We have added a new low magnification SEM image for the Cu mesh used in this study (Figure S4). The sizes of Cu wire (~100 μm) and pore (~150 μm) can be clearly seen from this image. A digital photo of the Cu mesh has also been added. In addition, several low magnification images of the Cu electrode after the reaction have been added to the revised supporting information (upper left of Figure S11, S12, and S14). Details about the size of Cu mesh have been added to the revised manuscript as follows:

“...we selected a cathode consisting of a copper mesh with large pores (average pore diameter of 150 μm and average Cu wire of 100 μm) and a pore density of 100 pores per inch (**Figure S4**).”

New Figure S4. Photograph (left) and low magnification SEM image (right) of the Cu mesh used in this study.

Potential effect of Fe and Ni cations from anode

To investigate the potential effects of Ni and Fe cations migrating from the anode side during alternating current operation, we have used IrO_x/Ti as an anode, as suggested by the reviewer. We observed a similar product distribution compared to that of reaction with the Ni foam anode (New Figure S16). We also intentionally added Ni and Fe cation impurities with varying concentrations to the catholyte. For both Ni and Fe, we observed detrimental effects when the metal cations were added (New Figure S17 and S18).

The following discussion has been added to the revised manuscript.

“To study the possible effect of impurities such as Ni and Fe ions, which could leach from the Ni foam anode and be transported to the cathode during alternating current operation⁵⁵, we performed ECR using stable IrO_x supported on titanium felt as the anode. This yielded similar CH_4 FEs compared to Ni-based anodes (**Figure S16**). We also intentionally added Ni and Fe ions to the catholyte with concentrations ranging from 0.1 to 1 ppm, and found that Ni and Fe impurities both show detrimental effects on CH_4 production (**Figure S17 and S18**). These results confirm that potential Ni and Fe impurities do not contribute to the high CH_4 FE obtained in our systems.”

New Figure S16. Performance with IrO_x/Ti anode. (a) CO₂RR product distribution as a function of current density using substitute anode IrO_x/Ti material. (b) Cell system voltage as a function of current density. To confirm that the improved selectivity for CH₄ was not due to impurities originating from the use of Ni mesh, we carried out additional experiments where we substituted the Ni mesh with an IrO_x/Ti anode^{1, 2}. We observed no significant difference from the use of an alternative anode material in our study. However, we do observe increased cell voltage. (Oxidation current density of 2.5 mA/cm² and oxidation time of 5 s; reduction time of 25 s were used for the CO₂ reduction at current density indicated above)

New Figure S17. Performance after adding Ni impurities. CO₂RR product distribution as a function of current density for (a) 0.1 ppm, (b) 0.5 ppm, (c) 1 ppm concentrations. To exclude the potential effects of Ni on the catalyst performance due to cation migration from the anode side

during our alternating current operation conditions we intentionally added Ni cations with different ppm concentrations to the catholyte. With the Ni cations, we observed detrimental effects on CH₄ selectivity at 1 ppm and 0.5 ppm concentrations, while the effect was less significant at 0.1 ppm. (Oxidation current density of 2.5 mA/cm² and oxidation time of 5 s; reduction time of 25 s were used for all conditions)

New Figure S18. Performance after adding Fe impurities. CO₂RR product distribution as a function of current density for (a) 0.1 ppm, (b) 0.5 ppm, (c) 1 ppm concentrations of Fe impurities. To exclude the potential effects of Fe on the catalyst performance due to cation migration from the anode side during our alternating current operation conditions we intentionally added Fe cations with different ppm concentrations to the catholyte. With the Fe cations, we observed severe detrimental effects on CH₄ selectivity at 1 and 0.5 ppm concentrations, and modest detrimental effects at 0.1 ppm. (Oxidation current density of 2.5 mA/cm² and oxidation time of 5 s; reduction time of 25 s were used for all conditions).

Effect of roughness factor:

To study the effect of surface roughness on product distribution in our system, we estimated the roughness factor of the Cu electrodes before and after reaction by measuring their double layer capacitance. We found that samples after both constant and alternating current operations exhibit relatively small roughness factors of 6 to 8. We also prepared a high surface area Cu mesh (with a roughness factor of 590) using a thermal treatment process. We found that high surface area Cu produced C₂H₄ as the main product while CH₄ FE was suppressed to below an FE of 1%. These

results are consistent with previous reports analyzing the effect of surface roughness on product distribution in electrochemical CO₂ conversion. We have added new data to the revised manuscript and discussed it in more detail in the next comment.

[#4] The results presented in Figs. 5 and 6 are nice because they demonstrate how the MEA system can be tuned to produce a high partial current of CH₄ over an extended period of time. What is missing though from the text is a clear explanation of why the system studies is selective to CH₄ vs C₂H₄, as one might expect.

We thank the review for this suggestion. As discussed in the revised manuscript, we have performed a series of additional experiments to study the effects of the Cu surface and alternating current operation on CH₄ production in our system. We found that alternating current operation not only activates the surface of Cu catalysts but is also critical to maintain high CH₄ selectivity. Without alternating current operation, activated Cu surface drops its high CH₄ selectivity very quickly (within minutes). We also performed additional experiments to confirm that Cu surfaces with small roughness factors favor the formation of CH₄ at high current density in our aqueous solution CO₂ electrolyzer. We have added the new data and discussion to the revised manuscript as follows:

“Origin of high CH₄ selectivity. Previous studies of ECR operating at low current densities of 10–50 mA/cm² in aqueous systems have suggested three key factors that affect hydrocarbon selectivity, including the nature of Cu active sites^{44, 49, 50}, catalyst surface roughness or loading⁵¹⁻⁵³, and the electrolyte⁵⁴. In our system, we found that catalyst morphology and alternating current operation are two crucial factors for high CH₄ production. To understand the effect of each factor, we performed a series of control experiments to decouple their roles. To investigate the contribution of operating mode, we performed ECR at a current density of 250 mA/cm² using alternating current until a stable CH₄ FE of over 70% was achieved for 1 hour (Figure 5a) and then switched the reaction to the constant current mode at the same current density. The CH₄ FE drops quickly to 20% after 1 hour of reaction while H₂ FE increases significantly. When the system was switched back to alternating operating mode again, CH₄ FE is recovered to the initial FE of over 70%.

These results confirm that even with oxidation-induced activated Cu surface, alternating current operation is needed to maintain high CH₄ FE in aqueous fed system at high current densities.

To study the possible effect of impurities such as Ni and Fe ions, which could leach from the Ni foam anode and be transported to the cathode during alternating current operation⁵⁵, we performed ECR using stable IrO_x supported on titanium felt as the anode. This yielded similar CH₄ FEs compared to Ni-based anodes (**Figure S16**). We also intentionally added Ni and Fe ions to the catholyte with concentrations ranging from 0.1 to 1 ppm, and found that Ni and Fe impurities both show detrimental effects on CH₄ production (**Figure S17 and S18**). These results confirm that potential Ni and Fe impurities do not contribute to the high CH₄ FE obtained in our systems.

New Figure 5. Roles of operation mode and surface roughness factor. (a) Product distribution of Cu mesh over time operated using alternating currents (oxidation current density of 2.5 mA/cm² and oxidation time of 5 s; reduction time of 25 s and CO₂ reduction current of 250 mA/cm²), followed by constant reduction current at 250 mA/cm² and then alternating currents. (b) Roughness factors of fresh Cu mesh, Cu mesh after constant reduction current operation, Cu mesh after alternating current operation, and thermally treated Cu mesh after alternating current operation. (c) Product distribution of thermally treated Cu mesh using alternating current operation. Samples after reaction were collected after being tested at 100, 250, 500, and 750 mA/cm² current densities for 40 minutes at each current density (total reaction time of 160 minutes). For samples with alternating current operations, the oxidation current density was 2.5 mA/cm², the oxidation and reduction times were 5 s and 25 s, respectively.

*To study the effect of surface morphology, we estimated the surface roughness factors of the catalysts from double-layer capacitance measurements. Cu electrodes after both constant and alternating current operations show similar roughness factors of 6-8 which is ca. 3 times higher than that of fresh Cu mesh (**Figure 5b** and **S19**). We reason that a relatively low surface roughness factor is the main reason for the high CH₄ FE observed in our study. This is consistent with what has been demonstrated in aqueous ECR systems using Cu mesh and Cu foils as electrodes, suggesting that a balance between surface roughness and applied potential is crucial for high CH₄ selectivity⁵³. To further confirm this point, we prepared a high surface area Cu mesh electrode using a thermal treatment process⁵⁶. With a high surface roughness factor of 590, the electrode produced C₂H₄ as the main hydrocarbon product (FE of 15 to 25%) while CH₄ production is suppressed (FE less than 1%) in current densities of 250 to 750 mA/cm² using alternating current operation (**Figure 5c** and **S20**). These results suggest that both low roughness factors and alternating current operation are needed to achieve high CH₄ FE and production rates in our aqueous solution ECR system.”*

In summary, the authors have presented some intriguing results; however, the interpretation of these results is currently incomplete and does not meet the standards expected for papers published in Nature Communications. A major revision of this work is needed for it to become acceptable.

We thank the reviewer for providing us with constructive comments and suggestions to improve the quality of the manuscript. With a substantial number of additional experiments, material characterizations and details, we unraveled the key factors governing our high CO₂-to-CH₄ performance in an aqueous solution electrolyzer. We believe that our work is now suitable for publication in Nature Communications.

Reviewer #2

The authors have investigated and developed a liquid-fed electrolytic system to convert CO₂ from an aqueous solution to CH₄. In the electrolysis, they employed mainly CO₂-saturated KHCO₃ aqueous solution and coupled the solution with an open matrix Cu mesh catalyst to enhance the supply of dissolved CO₂ molecules as the main CO₂ source to the cathode reaction site. The periodic oxidation treatment forms and keeps active porous Cu surface, which the authors think is important to keep the high Faradaic efficiency for CH₄ production (FE(CH₄)) of over 70% in a wide current density range. The aforementioned results and the authors' strategy to improve FE(CH₄) would be really interesting and significant in the research fields of both gas- and liquid-fed CO₂ electrolysis.

We thank the reviewer for these positive comments and other constructive comments below.

The present work, however, should have similar works previously reported by other research groups, which were conducted under similar experimental conditions. It would be necessary for the authors to clearly describe and discuss the difference between their present work and previous similar works and explain key factors to the good performance they achieved with quantitative scientific evidences. The reviewer requests the authors to respond to the following comments. (Reasons why the reviewer made these comments are represented in the attached reviewer report.)

We thank the reviewer for this suggestion. We have performed additional experiments, modelling and material characterizations to clearly point out the three simultaneously required factors for highly selective CO₂-to-CH₄ conversion at high current density from aqueous solutions. These three factors are:

- **Alternating current operation:** We now demonstrated in the revised manuscript that the alternating current operation not only activates the surface of Cu to generate methane selective surface but is also required to maintain the active surface. Without alternating current operation, the CH₄ FE drops rapidly for the previously activated Cu surface.

- **Dissolved and in-situ generated CO₂ sources:** Both experimental and modelling data in our study point out the importance of both CO₂ sources for CH₄ formation. At relatively low current densities, dissolved CO₂ is the main contributor to CO₂-to-CH₄ conversion while in-situ generated CO₂ plays an important role at high current densities.
- **Cu electrode with an open-matrix pore structure and a low roughness factor:** Our experimental and modelling data confirm the importance of both pore structure and roughness factor for high CH₄ selectivity at high current densities. The open matrix electrode effectively uses dissolved CO₂ as the carbon source which allows us to further optimize KHCO₃ concentration in the electrolyte to maximize CH₄ formation.

In the work that employed both dissolved and in-situ generated CO₂, (ACS Energy Lett., 7, 1712 (2022)), both catalyst structure and operational conditions are different from those used in our study. As discussed in our revised manuscript, the open matrix structure enables efficient dissolved CO₂ utilization, leading to high CH₄ partial current density. When only in-situ generated CO₂ is used (leading to much lower performance compared to when dissolved CO₂ is also used), our electrode shows a similar performance (CH₄ FE and partial current) to the electrode with dense structure reported (ACS Energy Lett., 7, 1712 (2022)).

In the revised manuscript, we have added the following discussions to compare our work and the reference 27:

“The highest recorded CH₄ FE was 27%, achieved at 500 mA/cm², corresponding to a CH₄ partial current density of 135 mA/cm² (Figure S21), which is comparable to previous reported data using in-situ generated CO₂ gas from bicarbonate electrolyte²⁷. These results confirm that the presence of dissolved CO₂ is crucial for high CH₄ FE and both CO₂ sources are required for high CH₄ selectivity at high current densities.”

“Compared to the dense electrodes used in previous studies²⁷, the open matrix catalyst used in this work effectively utilizes dissolved CO₂ as the carbon source. Therefore, highly concentrated KHCO₃ electrolyte is not needed to provide enough in-situ generated CO₂. As a result, the open

matrix electrode allows the utilization of KHCO₃ electrolyte with relatively low concentration, which produces favorable local pH for CO₂ conversion.”

Previous works on CO₂ reduction to CH₄ in aqueous systems often operated at relatively low current densities (10 -50 mA/cm²) and employed only dissolved CO₂ (Nat. Catal., 5, 259 (2022), Nat. Commun., 12:586 (2021), J. Mater. Chem. A, 10, 6187 (2022)). Therefore, it is not straightforward to compare those works with our study. However, these works have suggested the importance of the nature of active sites on CH₄ formation.

In the revised manuscript, we have added new results and discussions to investigate the origin of high CH₄ selectivity at high current densities in our work as follows:

*“Origin of high CH₄ selectivity. Previous studies of ECR operating at low current densities of 10–50 mA/cm² in aqueous systems have suggested three key factors that affect hydrocarbon selectivity, including the nature of Cu active sites^{44, 49, 50}, catalyst surface roughness or loading⁵¹⁻⁵³, and the electrolyte⁵⁴. In our system, we found that catalyst morphology and alternating current operation are two crucial factors for high CH₄ production. To understand the effect of each factor, we performed a series of control experiments to decouple their roles. To investigate the contribution of operating mode, we performed ECR at a current density of 250 mA/cm² using alternating current until a stable CH₄ FE of over 70% was achieved for 1 hour (**Figure 5a**) and then switched the reaction to the constant current mode at the same current density. The CH₄ FE drops quickly to 20% after 1 hour of reaction while H₂ FE increases significantly. When the system was switched back to alternating operating mode again, CH₄ FE is recovered to the initial FE of over 70%. These results confirm that even with oxidation-induced activated Cu surface, alternating current operation is needed to maintain high CH₄ FE in aqueous fed system at high current densities.*

*To study the possible effect of impurities such as Ni and Fe ions, which could leach from the Ni foam anode and be transported to the cathode during alternating current operation⁵⁵, we performed ECR using stable IrO_x supported on titanium felt as the anode. This yielded similar CH₄ FEs compared to Ni-based anodes (**Figure S16**). We also intentionally added Ni and Fe ions to the catholyte with concentrations ranging from 0.1 to 1 ppm, and found that Ni and Fe impurities*

both show detrimental effects on CH_4 production (**Figure S17 and S18**). These results confirm that potential Ni and Fe impurities do not contribute to the high CH_4 FE obtained in our systems.

New Figure 5. Roles of operation mode and surface roughness factor. (a) Product distribution of Cu mesh over time operated using alternating currents (oxidation current density of $2.5 \text{ mA}/\text{cm}^2$ and oxidation time of 5 s; reduction time of 25 s and CO_2 reduction current of $250 \text{ mA}/\text{cm}^2$), followed by constant reduction current at $250 \text{ mA}/\text{cm}^2$ and then alternating currents. **(b)** Roughness factors of fresh Cu mesh, Cu mesh after constant reduction current operation, Cu mesh after alternating current operation, and thermally treated Cu mesh after alternating current operation. **(c)** Product distribution of thermally treated Cu mesh using alternating current operation. Samples after reaction were collected after being tested at 100, 250, 500, and 750

mA/cm² current densities for 40 minutes at each current density (total reaction time of 160 minutes). For samples with alternating current operations, the oxidation current density was 2.5 mA/cm², the oxidation and reduction times were 5 s and 25 s, respectively.

*To study the effect of surface morphology, we estimated the surface roughness factors of the catalysts from double-layer capacitance measurements. Cu electrodes after both constant and alternating current operations show similar roughness factors of 6-8 which is ca. 3 times higher than that of fresh Cu mesh (**Figure 5b** and **S19**). We reason that a relatively low surface roughness factor is the main reason for the high CH₄ FE observed in our study. This is consistent with what has been demonstrated in aqueous ECR systems using Cu mesh and Cu foils as electrodes, suggesting that a balance between surface roughness and applied potential is crucial for high CH₄ selectivity⁵³. To further confirm this point, we prepared a high surface area Cu mesh electrode using a thermal treatment process⁵⁶. With a high surface roughness factor of 590, the electrode produced C₂H₄ as the main hydrocarbon product (FE of 15 to 25%) while CH₄ production is suppressed (FE less than 1%) in current densities of 250 to 750 mA/cm² using alternating current operation (**Figure 5c** and **S20**). These results suggest that both low roughness factors and alternating current operation are needed to achieve high CH₄ FE and production rates in our aqueous solution ECR system.”*

[#1] Please describe the details of the multiphysics modeling performed in this work (all of the parameters and their values used in the modeling), which are necessary for readers to fully understand the situation and reaction environment the authors consider and to compare the simulation results.

We thank the reviewer for pointing out specific omissions regarding the modeling description. As described in Response 1 to Reviewer #1, additional detail has been included regarding the multiphysics modeling. These additions include clarifying and adding more detail to **Figure 1**, adding updated description in the “Modelling reaction environment and CO₂ availability” sub-section in the “Results” section, the “Multiphysics modeling” sub-section in the “Methods” section, and the bulk bicarbonate concentrations in **Table S2** of the Supplementary Information. In regard to the specific comment about CO₂ mole fraction as compared to Lees et al. (Ref. 27),

upon comparison of our **Figure S2** with their **Figure S12**, we note that the dissolved CO₂ concentration in their work is somewhat higher in the membrane region, and somewhat lower in near the flow-plate interface. This makes sense: they consider higher concentration of catholyte (3M KHCO₃ compared to our 0.3M KHCO₃), but with lower bulk CO₂ concentration (N₂ sparged vs CO₂ sparged).

[#2] The following description seen in the "Modeling the reaction environment and CO₂ availability" Subsection (p. 4) should be revised to avoid misunderstanding, for example, by clearly explaining that the reaction environment considered in the modeling and in the electrolysis of this work is different from that considered in previous works like Ref. 27; "State-of-the-art bicarbonate electrolyzers rely on catalysts within a dense porous matrix with the CO₂ source in-situ generation from bicarbonate. These electrolyzers often suffer from low CH₄ FE at industrially relevant current densities, as predicted in our simulations (Figure 1b)".

We agree with the Reviewer's concerns. We removed the sentence:

These electrolyzers often suffer from low CH₄ FE at industrially relevant current densities, as predicted in our simulations (Figure 1b).

And we changed:

To understand the cause of the reduced performance at higher current density

To:

Due to the critical importance of CO₂ availability on electrolyzer performance

[#3] Please examine the validity of the simulation results by comparing them with corresponding experimental results.

Direct comparison of the modeled results with experimental results are now included in the Supplementary Information, **Figure S23 and S24**, reproduced below. We observe that the simulation replicates all major trends seen in experiments for all cases tested, including the relative performance comparing AEM vs BPM, CO₂ vs N₂ sparging, and the different electrolyte

concentrations, with the possible exception of the modeled overestimate of the 1M KHCO_3 at higher current densities. It is not entirely clear why there is a minor mismatch in the trends here, but we suspect it is related to the CO_2 bubbling physics, which are quite complex, the detailed treatment of which is somewhat outside the scope of this work. The generally good matching gives further confidence in the results and conclusions presented in Figure 1.

The following discussion has been added to the revised manuscript:

“To validate the experimental trends and the conclusions from the multiphysics model, the model was tested against various experimental configurations, including AEM operation (Figure S23), various sparging gases (Figure S23), and at different electrolyte concentrations (Figure S24). The model can replicate cell performance trends in all cases.”

See Figure S23. Comparison of experimental and modelled ECR performance for various configurations. Experimental CH_4 FE compared with predicted FE from multiphysics modelling for CO_2 and N_2 sparging with a BPM and with an AEM with CO_2 sparging. For the experiments, the oxidation current density was 2.5 mA/cm^2 and the oxidation and reduction times were 5 s and 25 s, respectively. Catholyte is 0.3M KHCO_3 in experiments and modelling.

New Figure S24 Comparison of experimental and modelled ECR performance for various catholyte concentrations. Experimental CH₄ FE compared with predicted FE from multiphysics modelling for various catholyte (KHCO₃) concentrations. For the experiments, the oxidation current density was 2.5 mA/cm² and the oxidation and reduction times were 5 s and 25 s, respectively. In all cases a BPM with CO₂ sparging is used.

[#4] Please clearly express the details of the electrolysis cell and the electrolysis system employed in this work.

We have added an additional figure that includes a schematic photo, reactor configuration and dimensional drawing, as shown in Figure S5. The following discussion has been added to the revised manuscript:

“Details of the electrolysis cell and the electrolysis system employed in this work are described in Figure S5.”

New Figure S5. Electrochemical testing system and detailed flow channel design. (a) System set-up of electrochemical CO₂ conversion from aqueous solutions to hydrocarbons. (b) Schematic illustration of the electrolysis cell. From left to right: Cathode flow plate, PTFE spacer, copper mesh, ion exchange membrane, PTFE gasket, nickel foam, anode flow plate. (c) The cell dimensional drawing. All dimensions are in inches.

[#5] Please show and compare SEM images of different domains at the surface of Cu mesh whose appearances are quite different before and after the CO₂ electrolysis. Also, the authors should quantitatively express the surface states (especially the porous feature) of the Cu mesh before and after the CO₂ electrolysis, for example, by using surface area and surface roughness.

SEM images of different domains at the surface of Cu mesh

As suggested by the reviewer, we have characterized the samples before and after reactions at different domains of Cu surface. Additional SEM images taken at different domains have been added to the Supplementary Information, as Figure S11, S12, and S14.

New Figure S11. SEM images of fresh Cu mesh from multiple locations.

New Figure S12. SEM images of Cu mesh after constant reduction current operation from multiple locations. The sample for SEM was collected after being tested at 100, 250, 500, and 750 mA/cm² current densities for 40 minutes at each current density (total reaction time of 160 minutes).

Figure S14. SEM images of Cu mesh after alternating current operation from multiple locations. The sample for SEM was collected after being tested at 100, 250, 500, and 750 mA/cm² current densities for 40 minutes at each current density (total reaction time of 160 minutes). The oxidation current density was 2.5 mA/cm². The oxidation and reduction times were 5 s and 5 s, respectively.

Analysis of surface roughness factor

As suggested by the reviewer, we have measured the electrochemical double-layer capacitance of the Cu mesh before and after the CO₂ electrolysis to quantitatively compare the change in the surface states. We have plotted the data and added it to the revised manuscript and Supplementary Information, as shown in Figure 5, S19 and S20. We have revised the methods section and added text as follows:

“Electrochemical double-layer capacitance measurement

Electrochemical double-layer capacitance was determined by measuring cyclic voltammetry (CV) in a H-cell, made up of two different chambers and separated by an anion exchange membrane. A three-electrode setup was used, comprised of the working electrode and a KCl saturated reference electrode (Ag/AgCl electrode) in the cathode chamber and a platinum counter electrode placed in the anode chamber. All potentials for this measurement were determined against the Ag/AgCl reference electrode. The dimension of the working electrode which was immersed in the electrolyte is 1 cm x 1cm. First, the potential range of the non-Faradaic current regime was determined from CV. CV measurements were then conducted in a CO₂ saturated 0.3 M KHCO₃ electrolyte solution by sweeping the potential in the non-faradaic region between -0.55 and -0.65 V vs. Ag/AgCl with 3 cycles for each scan rate of 0.1, 0.2, 0.4, 0.6 0.8, 1, 1.2, and 1.4 V/s. For the thermally treated Cu mesh, scan rates of 0.005, 0.01, 0.02, 0.04, and 0.06 V/s were used. Roughness factors of Cu electrodes were estimated by comparing their capacitances against that of an ideally smooth Cu surface (0.029 mF/cm²)⁵⁶.”

The discussion of the effect of the surface roughness factors on the CH₄ production in our system has been added to the revised manuscript as follows:

*“To study the effect of surface morphology, we estimated the surface roughness factors of the catalysts from double-layer capacitance measurements. Cu electrodes after both constant and alternating current operations show similar roughness factors of 6-8 which is ca. 3 times higher than that of fresh Cu mesh (**Figure 5b** and **S19**). We reason that a relatively low surface roughness factor is the main reason for the high CH₄ FE observed in our study. This is consistent with what*

*has been demonstrated in aqueous ECR systems using Cu mesh and Cu foils as electrodes, suggesting that a balance between surface roughness and applied potential is crucial for high CH₄ selectivity⁵³. To further confirm this point, we prepared a high surface area Cu mesh electrode using a thermal treatment process⁵⁶. With a high surface roughness factor of 590, the electrode produced C₂H₄ as the main hydrocarbon product (FE of 15 to 25%) while CH₄ production is suppressed (FE less than 1%) in current densities of 250 to 750 mA/cm² using alternating current operation (**Figure 5c** and **S20**). These results suggest that both low roughness factors and alternating current operation are needed to achieve high CH₄ FE and production rates in our aqueous solution ECR system.”*

New Figure S19. Electrochemical double-layer capacitance measurements. Cyclic voltammograms in the non-faradaic region for (a) Cu mesh before electrolysis, (b) Cu mesh after electrolysis under constant reduction current, and (c) Cu mesh after electrolysis under alternating 25 s reduction - 5 s oxidation currents. (d) Charging current density plotted as a function of scan rate. The samples (b and c) were collected after being tested at 100, 250, 500, and 750 mA/cm² current densities for 40 minutes at each current density (total reaction time of 160 minutes). For sample (c), the oxidation current density was 2.5 mA/cm². The oxidation and reduction times were 5 s and 25 s, respectively.

New Figure S20. Electrochemical double-layer capacitance measurements of thermally treated Cu mesh. (a) cyclic voltammograms in the non-faradaic region for Cu mesh after thermal treatment and electrolysis under alternating 25 s reduction – 5 s oxidation currents, and (b) the charging current density plotted as a function of scan rate. The sample was collected after being tested at 100, 250, 500, and 750 mA/cm² current densities for 40 minutes at each current density (total reaction time of 160 minutes). The oxidation current density was 2.5 mA/cm². The oxidation and reduction times were 5 s and 25 s, respectively.

[#6] Please verify and show the reproducibility of the phenomenon that Cu nanoparticles are evolved under a constant-current condition and also explain driving forces to evolve Cu nanoparticles on the surface in a reducing environment.

Reproducibility of the Cu nanoparticle formation.

We carried out multiple trials (3) of the continuous electrolysis using a constant current electrolysis and characterized the sample using SEM. All samples show the presence of nanoparticles with irregular shapes. We have added this result to the revised supporting information as Figure S13.

New Figure S13. SEM images of Cu mesh after constant reduction current operation for three different trials. The sample for SEM was collected after being tested at 100, 250, 500, and 750 mA/cm² current densities for 40 minutes at each current density (total reaction time of 160 minutes).

Explain the driving force responsible for the evolution of Cu nanoparticles.

Previous studies using operando characterizations of catalysts under CO₂ reduction conditions suggested that the Cu surface is dynamic and undergoes significant morphological changes due to multiple processes including dissolution/redeposition, agglomeration, fragmentation and reshaping. As the surface reconstruction is accelerated at higher reaction rates, we would expect significant changes on Cu surface in our testing conditions (current densities in the range of 250-750 mA/cm²).

We have added the discussion about surface reconstruction to the revised manuscript as follows:

“We found that CO₂ reduction conditions have a significant impact on the surface of Cu catalysts. SEM characterizations after ECR reactions with constant reduction current reveal the formation of Cu nanoparticles with irregular shapes (Figure 4b, S12 and S13) as opposed to a smooth surface of the sample before reaction (Figure 4a and S11). This observation is consistent across different locations of multiple samples. Previous studies using operando characterizations of catalysts under ECR conditions suggested that the Cu surface is dynamic and undergoes significant morphological changes due to multiple processes, including dissolution/redeposition, agglomeration, fragmentation and reshaping^{33, 34, 46-48}. Formation of Cu nanoparticles on the Cu surface during ECR has been observed in a previous study at a current density below 50 mA/cm²

Ref(12). Because the surface changes are accelerated at higher reaction rates or more negative applied potentials³³, formation of Cu nanoparticles at current densities of 500 to 750 mA/cm² in our system is reasonable. In sharp contrast to the constant reduction operation, alternating reduction-oxidation currents were found to induce the formation of a porous Cu surface (Figure 4c). The porous structure becomes more pronounced when the reduction time is reduced from 50 s to 5 s (Figure 4d - 4f and S14), suggesting that oxidation cycle frequency is an important factor governing the morphology of the catalyst.”

In addition to this discussion, we have also revised Figure 4 by adding characterization of Cu surface with different oxidation reduction operating conditions.

Revised Figure 4. Surface changes induced by alternating reduction-oxidation current. Scanning electron microscopy (SEM) images of Cu mesh electrode before and after CO₂ reduction using fixed reduction current (**a** and **b**); and after alternating reduction-oxidation times of 25 s - 5 s, 50 s - 5 s, 15 s - 5 s, and 5 s - 5 s (**c**, **d**, **e**, and **f**). Cu 2p X-ray photoelectron spectroscopy (XPS) spectra of Cu mesh electrode after CO₂ reduction using fixed currents (**g**); and alternating 25 s - 5 s reduction-oxidation current (**h**). The samples for SEM and XPS were collected after being tested at 100, 250, 500, and 750 mA/cm² reduction current densities for 40 minutes at each current regime (total reaction time of 160 minutes). The oxidation current density and time were

2.5 mA/cm² and 5 s, respectively. The reduction time considered for samples (c, d, e, and f) are 25 s, 50 s, 15 s, and 5 s, respectively. Scale bar in a-f = 200 nm.

[#7] Please clearly describe standard electrolysis conditions in this work (e.g., the concentration of K⁺ ion in the catholyte), electrolysis duration for one current density condition, and how to conduct online gas chromatography (GC) analysis (e.g., the method to introduce gaseous products in the electrolysis to the GC instrument, the flow rate of carrier gas if used).

We thank the reviewer for his comment. We have added Figure S5, which includes a schematic photo, reactor configuration and dimensional drawing. The following discussion has been added to the revised manuscript 'Methods':

"In the MEA flow cell, copper mesh (100 pores per inch (ppi)) and nickel foam (1.5 mm thickness; 80-100 ppi, MTI Corp.) were used as the cathode and anode, respectively, and were separated by a bipolar membrane (Fumasep) or an anion exchange membrane (Fumasep). The exposed sizes of both the cathode and anode electrodes were 1 cm x 2 cm (a geometric area of 2 cm² was used for all current density calculations). The copper mesh was configured to be in direct contact with the membrane. The polytetrafluoroethylene (PTFE) gasket was used to prevent contact between anode and cathode titanium flow plates. A PTFE spacer was used to avoid direct contact between the Cu mesh and the cathode flow plate.

For the electrolysis system, a potentiostat (Autolab PGSTAT204) with a current booster (Metrohm Autolab, 10A) was used for all experiments. Two peristaltic pumps were used to circulate the anolyte (1 M KOH; 200 mL) and catholyte (0.1M, 0.3 M, or 1 M KHCO₃; 1000 mL) between the reservoirs and the electrochemical cell at a flow rate of 30 mL/min. Before the measurement, CO₂ gas (Praxair, 99.99%) or N₂ gas (Praxair, 99.99%) was purged in the KHCO₃ aqueous solution for at least 30 min. CO₂ was continuously purged into the catholyte throughout the experimental process (during both oxidation and reduction cycles) at a flow rate of 50 standard cubic centimeters per minute (sccm). Electrolysis was carried out for 2500 s at each tested current density, with the reaction gaseous product being analyzed every 15 min. Gas products coming out from the cell was carried by the CO₂ or N₂ stream bubbled through the catholyte reservoir to a 6-

way valve where it was injected to a gas chromatography (GC, PerkinElmer Clarus 590) for quantification. The GC is equipped with a thermal conductivity detector (TCD) operated at 180 °C and a flame ionization detector (FID) operated at 250 °C. A molecular sieve (5A) packed column (Supelco) connected to the TCD was used to analyze CO, CO₂ and H₂ products while a Carboxen-1000 packed column (Supelco) connected to the FID was employed to quantify CH₄, C₂H₄ and other potential hydrocarbons. Both columns were operated at a fixed temperature of 180 °C and with Ar carrier gas (flowrate of 20 sccm).”

[#8] Please clearly describe the definition of the total current considered in this work. The value of the total current is important and affects Faradaic efficiency and energy efficiency reported in this work. The definition of Faradaic efficiency used in this work should be reviewed on the basis of the following comment. Not only the reduction current but also the oxidation current contributes to total electricity consumption. Thus, Faradaic efficiency (or coulombic efficiency) should be the molar ratio of produced CH₄ multiplied by the number of electron consumed (= 8) to electrons flowing during the entire reduction-oxidation operation. From this point of view, the reviewer feels that it would be easy to estimate the quantity of electricity during the reduction-oxidation operation while it would not be simple to estimate (weighted average) total current.

In this work, we consider only the charge transferred during the reduction cycle for the Faradaic efficiency (FE) calculation. Thus, only reduction current is considered for the selectivity calculation. We have revised the manuscript to make this point clear as discussed below.

There are two reasons we only consider charge transferred during the reduction cycle for the FE calculation. First, this will provide information about the intrinsic selectivity of the catalysts which can be used to compare with reported data, including data from some studies using pulse electrolysis. Second, the charge transferred during the oxidation cycles is typically several hundred times smaller than that in the reduction cycle due to the very small oxidation current and short oxidation time. Thus, including oxidation charge will not noticeably change the FE values. We have recalculated the FE and energy efficiency (EE) of the three conditions we reported in the comparison table (Table S1). The new data are compared with the old data in the table below:

Reduction and oxidation conditions	Reduction current (mA/cm ²)	CH ₄ FE without oxidation charges	CH ₄ FE with oxidation charges
Reduction: 25s	250	72.9	72.75
Oxidation current: 2.5 mA/cm ²	500	74.3	74.22
	750	67.8	67.75
Oxidation time: 5s			

As seen from the table above, the differences between with and without oxidation charges are very small. Nevertheless, we have added new FE values with oxidation charges included to the revised Table S1.

We consider the factor $\frac{\Delta t_r}{\Delta t_r + \Delta t_o}$ as weighted effective operation time to calculate effective CH₄ partial current density based on a practical perspective. While the oxidation cycle does not appreciably contribute to the total electrical energy consumption as discussed above, CH₄ is not produced during this cycle. Thus, the oxidation time can be practically considered as down time of the electrolyzer. From this perspective, using effective CH₄ partial current with weighted operation time provides a fair comparison to reported data using constant current operation modes. We have added these details to the revised manuscript:

“where $J_{CH_4,eff}$ is the effective CH₄ current density. The effective current density reflects the effective operation time of the electrolyzer considering both the time for CO₂ conversion (reduction) and the time for catalyst regeneration (oxidation). Δt_r and Δt_o are the reduction and oxidation time in each reduction-oxidation cycle. The factor $\frac{\Delta t_r}{\Delta t_r + \Delta t_o}$ is used in the expression to weight operation (conversion) time to the total time.”

Regarding the factor $\frac{\Delta t_r}{\Delta t_r + \Delta t_o}$ in the Faradaic efficiency formula, we do not use it to calculate weighted current. Instead, this factor was used to calculate the concentration of gas products in the gas outlet stream. Because the reduction and oxidation cycles are relatively short and the CO₂ is

constantly flowed through the system (during both reduction and oxidation cycles), gas produced during the reduction cycle is diluted by CO₂ flow during the oxidation cycle. Thus, the actual gas molar fraction during the reduction cycle used for FE calculation is calculated as $x'_k \cdot \frac{\Delta t_r + \Delta t_o}{\Delta t_r}$ where x'_k is the diluted molar fraction (mixing of gas production in both oxidation and reduction cycle) measured by the Gas chromatography.

We have revised the formula for the FE calculation and added the above explanation to the revised manuscript as follows:

“The FE of ECR gas products were calculated as follows:

$$FE_k = \frac{n_k \cdot F \cdot x_k \cdot F_m}{I} \times 100\%$$

Where FE_k is the FE of product k , n_k is the number of electrons (for $k = CH_4$, n_k is 8) transferred to form gaseous product k , F is Faraday’s constant ($96,485 \text{ C mol}^{-1}$), F_m is the molar flow rate of the gas outlet stream in $\frac{\text{mol}}{\text{s}}$. x_k is the molar fraction of the gas product k in the gas outlet stream during reduction cycle. I is the total (applied) current in Amperes (A) during reduction cycle.

Because the reduction and oxidation cycles are relatively short and the CO₂ is constantly flowed through the system (during both reduction and oxidation cycles), gas produced during the reduction cycle is diluted by CO₂ flow during the oxidation cycle. Thus, the actual molar fraction of the gas product during the reduction cycle is calculated as follows:

$$x_k = x'_k \cdot \frac{\Delta t_r + \Delta t_o}{\Delta t_r}$$

where x'_k is the diluted concentration (mixing of gas production in both oxidation and reduction cycle) measured by the Gas chromatography. Δt_r and Δt_o are the reduction and oxidation time in each reduction-oxidation cycle.”

[Additional comment] In this work, not only CO₂ reduction and hydrogen evolution reaction but also hydrogen oxidation reaction and oxygen evolution reaction during oxidation and oxygen reduction reaction during reduction could occur in the alternating reduction-oxidation operation.

The absolute value of the oxidation current is much smaller than that of the reduction current, and the period of the oxidation is shorter than that of the reduction. Hence, the contribution of hydrogen oxidation (consumption) to decreasing FE(H₂) might be small. However, the reviewer recommends the authors to mention the possibilities that redox reactions of byproducts (hydrogen and oxygen) might occur.

We thank the reviewer for this insight. We agree that hydrogen oxidation, leading to the consumption of H₂ in the outlet stream, could lead to a lower H₂ FE. However, as the reviewer pointed out, the oxidation current is relatively small, and the oxidation time is shorter than the reduction time. We would expect an insignificant contribution of H₂ oxidation. For example, in one of our test conditions (reduction current 250 mA/cm²; reduction time: 25s (total reduction charge of 6.25 C/cm²); oxidation current and time: 2.5 mA/cm² and 5s (total oxidation charge of 1.25 x 10⁻² C/cm²), the FE for H₂ is 19.39. If we assume all oxidation current goes to H₂ oxidation current, then the FE of H₂ is reduced to 19.15.

As suggested by the reviewer, we have added this discussion to the revised manuscript:

“During the oxidation cycle, H₂ produced from the reduction cycle may be oxidized, contributing to the observed low H₂ FE. However, because the number of charges during the oxidation cycle (0.0015 C/cm²) is much smaller than those in the reduction reaction (6.25 C/cm²), this possible contribution is insignificant.”

REVIEWER COMMENTS

Reviewer #1 (Remarks to the Author):

The authors have done good job of addressing the comments arising from the first review of this work. Not addressed though, is the questions of why the only product of CO₂RR is CH₄, and no other products are observed with any significant FE. The only response given by the author is their observation that the roughness of their Cu samples is low (~6-8) and that they use pulsed electrolysis. These are operational responses but ones that provide any chemical perspective. Therefore, it is appropriate to ask what is it about the authors experimental set up at leads to up to a 70% FE for CH₄. In other words what is the pH near the Cu surface; is the source of H atoms for the reduction of CO₂, H⁺ or H₂O. These are important question that remain unanswered.

Response to the reviewer:

The authors have done good job of addressing the comments arising from the first review of this work. Not addressed though, is the questions of why the only product of CO₂RR is CH₄, and no other products are observed with any significant FE. The only response given by the author is their observation that the roughness of their Cu samples is low (~6-8) and that they use pulsed electrolysis. These are operational responses but ones that provide any chemical perspective. Therefore, it is appropriate to ask what is it about the authors experimental set up that leads to up to a 70% FE for CH₄. In other words what is the pH near the Cu surface; is the source of H atoms for the reduction of CO₂, H⁺ or H₂O. These are important question that remain unanswered.

Response:

We thank the reviewer for this suggestion. We agree that local surface pH plays a critical role in product distribution on Cu surface. Previous work (Nature Communications, 2018, 9, 925) highlighted the importance of the balance between surface roughness, local pH and CO₂ availability for the production of C1 and C2 compounds. As suggested by the reviewer, we have performed numerical simulations to study the effect of local pH in the catalyst layer. We found that as the current density increases from 100 to 500 mA/cm², the local pH ranges from 10 to 12 throughout the catalyst layer (**Figure S21**). This result confirms that the proton source for CO₂ reduction originates from surface water molecules. Our results also show that the local pH is in the range from 10 to 11 at the two sides of the catalyst layer, where the vast majority of CO₂ conversion is expected to occur due to the locally higher CO₂ concentration predicted (**Figure S1 and S2**).

Our experimental data further support the role of pH within the catalyst layer. Our results show that using lower KHCO₃ catholyte concentration leads to a higher ethylene selectivity (**Figure S22**). Lower KHCO₃ concentrations have reduced buffering effects, presumably leading to higher local pH, outside of the optimal pH range for CH₄ formation. These results are consistent with the abovementioned report, showing that a relatively flat Cu surface (roughness factor of 1.3) produces a high CH₄ selectivity of around 60% as the local surface pH stays around 10.5 – 11 (*Nature Communications, 2018, 9, 925*).

We have added the discussion below into the revised manuscript:

“Finally, the local pH in the catalyst layer, estimated from numerical simulation for the 0.3 M KHCO_3 catholyte, is between 10 and 12 at current densities of 100 – 500 mA/cm^2 . This suggests that water proton source for CH_4 formation (**Figure S21**). Our results are consistent with previous works showing an optimal local pH of 10.5 - 11 for CH_4 formation on a relatively flat Cu surface.⁵³ Experiments using lower KHCO_3 catholyte concentration show a higher $\text{C}_2\text{H}_4:\text{CH}_4$ ratio (**Figure S22**), further supporting the role of pH within the catalyst layer to tune hydrocarbon selectivity. Lower KHCO_3 concentrations have reduced buffering effects, leading to higher local pH near the flow channel (**Figure S21**), presumably outside of the optimal pH range for CH_4 formation.”

New Figure S21. Local pH within the catalyst layer. Simulated pH for current densities at 100 mA/cm^2 , 250 mA/cm^2 , and 500 mA/cm^2 using the open matrix catalyst with a BPM and 0.3M KHCO_3 catholyte with CO_2 sparging.

New Figure S22. Effect of electrolyte concentrations. Gas product distribution at different current densities of Cu mesh operated using alternating negative and positive currents and 0.1 M KHCO₃ (a); 0.3 M KHCO₃ (b); and 1 M KHCO₃ (c). Effect of electrolyte concentration on C₂H₄:CH₄ ratio (d). The oxidation current density was 2.5 mA/cm². The oxidation and reduction times were 5 s and 25 s, respectively.

REVIEWERS' COMMENTS

Reviewer #1 (Remarks to the Author):

The authors have now addressed the last question posed by the reviewer. The work can now be accepted for publication.

Reviewer #1

The authors have now addressed the last question posed by the reviewer. The work can now be accepted for publication.

We thank the reviewer for supporting our work.